# A hierarchy of timescales explains distinct effects of local inhibition of primary visual cortex and frontal eye fields

Luca Cocchi[1,2]*, Martin V Sale[1], Leonardo L Gollo[2], Peter T Bell[1], Vinh T Nguyen[2], Andrew Zalesky[3], Michael Breakspear[2,4], Jason B Mattingley[1,5]

[1]Queensland Brain Institute, The University of Queensland, Brisbane, Australia; [2]QIMR Berghofer Medical Research Institute, Brisbane, Australia; [3]Melbourne Neuropsychiatry Centre, The University of Melbourne, Melbourne, Australia; [4]Metro North Mental Health Service, Brisbane, Australia; [5]School of Psychology, The University of Queensland, Brisbane, Australia

**Abstract** Within the primate visual system, areas at lower levels of the cortical hierarchy process basic visual features, whereas those at higher levels, such as the frontal eye fields (FEF), are thought to modulate sensory processes via feedback connections. Despite these functional exchanges during perception, there is little shared activity between early and late visual regions at rest. How interactions emerge between regions encompassing distinct levels of the visual hierarchy remains unknown. Here we combined neuroimaging, non-invasive cortical stimulation and computational modelling to characterize changes in functional interactions across widespread neural networks before and after local inhibition of primary visual cortex or FEF. We found that stimulation of early visual cortex selectively increased feedforward interactions with FEF and extrastriate visual areas, whereas identical stimulation of the FEF decreased feedback interactions with early visual areas. Computational modelling suggests that these opposing effects reflect a fast-slow timescale hierarchy from sensory to association areas.

**\*For correspondence:** Luca. Cocchi@qimrberghofer.edu.au

**Competing interests:** The authors declare that no competing interests exist.

## Introduction

The primate visual system is a hierarchical network of feedforward and feedback connections that supports visual perception, object recognition and selective attention (*Gilbert, 2013*). At the earliest level of the visual hierarchy, the primary visual cortex receives input from the lateral geniculate nucleus and propagates this information to higher association areas including the posterior parietal cortex, ventral temporal cortex and frontal eye fields (FEF) (*Hubel and Wiesel, 1962*; *Felleman and Van Essen, 1991*; *Van Essen and Maunsell, 1983*; *Maunsell and van Essen, 1983*). During active perception, neural signals originating in these later cortical areas provide top-down modulation of activity in early visual areas, depending on factors such as expectation and context (*Gilbert, 2013*; *Felleman and Van Essen, 1991*; *Moore and Armstrong, 2003*; *Bastos et al., 2015*).

Functional interactions between early sensory regions and later association areas have been investigated in humans using task-based functional magnetic resonance imaging (fMRI) (*Bressler et al., 2008*; *Vossel et al., 2012*). This work has demonstrated that task demands play an important role in determining the nature of the interactions between early and late visual cortex (*Gilbert, 2013*). For example, studies combining fMRI with transcranial magnetic stimulation (TMS) during performance of visual tasks have shown that stimulation of the FEF can modulate neural activity in several visual areas, including the primary visual cortex (area V1) (*Bressler et al., 2008*; *Ruff et al., 2006*, *2008*). Increments in the strength of excitatory TMS over the right FEF have been shown to increase

**eLife digest** In humans, the parts of the brain involved in vision are organized into distinct regions that are arranged into a hierarchy. Each of these regions contains neurons that are specialized for a particular role, such as responding to shape, color or motion. To actually 'see' an object, these different regions must communicate with each other and exchange information via connections between lower and higher levels of the hierarchy. However, it remains unclear how these connections work.

A brain region called the primary visual cortex is the lowest level of the visual cortical hierarchy as it is the first area to receive information from the eye. This region then passes information to higher regions in the hierarchy including the frontal eye fields (FEF), which help to control visual attention and eye movements. In turn, the FEF is thought to provide 'feedback' to the primary visual cortex.

Cocchi et al. examined how the FEF and primary visual cortex communicate with the rest of the brain by temporarily inhibiting the activity of these regions in human volunteers. The experiments show that inhibiting the primary visual cortex increased communication between this region and higher level visual areas. On the other hand, inhibiting the FEF reduced communication between this region and lower visual areas.

Computer simulations revealed that inhibiting particular brain regions alters communication between visual regions by changing the timing of local neural activity. In the simulations, inhibiting the primary visual cortex slows down neural activity in that region, leading to better communication with higher regions, which already operate on slower timescales. By contrast, inhibition of the FEF reduces its influence on lower visual regions by increasing the difference in timescales of neural activity between these regions.

The next step is to determine whether similar mechanisms regulate changes in the activity of neural networks outside of the visual system.

neural activity in early retinotopic cortex representing the peripheral visual field, and to reduce activity in central visual-field representations (*Ruff et al., 2006*). This neurophysiological effect confers a perceptual advantage for detection of stimuli in the visual periphery relative to the fovea (*Ruff et al., 2006*).

Despite evidence for dynamic bidirectional interactions between FEF and early visual cortex during task-related visual processing, numerous findings from resting-state fMRI (rsfMRI) studies, in both human and non-human primates, suggest there is little or no functional coupling between these regions in the absence of active task demands (i.e., in the resting-state) (*Vincent et al., 2007*; *Belcher et al., 2013*; *Damoiseaux et al., 2006*; *Yeo et al., 2011*; *Power et al., 2011*; *Mantini et al., 2013*; *Gordon et al., 2016*). At rest, FEF belongs to extensive networks of fronto-parietal regions (*Yeo et al., 2011*; *Power et al., 2011*), whereas the early visual cortex, including area V1, belongs to a primary visual network encompassing occipital and inferior parietal regions (*Yeo et al., 2011*; *Power et al., 2011*; *Gordon et al., 2016*). Despite this apparent functional segregation between visual networks encompassing early visual cortex and FEF at rest, the nature of any latent interactions between them remains unknown. Likewise, the neural principles facilitating the emergence of integration between segregated regions at opposing ends of the visual cortical hierarchy remain unclear.

Here we combined rsfMRI and non-invasive brain stimulation to examine the causal influence of perturbations of local neural activity within early and late visual areas – specifically areas V1/V2 and FEF – in the absence of visual task demands. Across two separate imaging sessions, we employed continuous theta-burst TMS (*Huang et al., 2005*) to inhibit intrinsic neural activity either within the right occipital pole or within the right FEF. We recorded resting-state brain activity immediately before and after TMS, and examined the influence of this perturbation on functional and effective connectivity between the targeted regions and the rest of the brain. We also employed computational modelling to provide candidate mechanisms for expected changes in interactions between cortical regions following local stimulation. Based on recent empirical (*Hasson et al., 2008*; *Murray et al., 2014*; *Bassett et al., 2013*; *Lerner et al., 2011*; *Honey et al., 2012*; *Gauthier et al.,*

*2012*) and computational (*Gollo et al., 2015*; *Chaudhuri et al., 2015*) work we tested the hypothesis that a temporal hierarchy of timescales, recapitulating the hierarchical organization of neuronal receptive fields, can explain the effects of local stimulation on inter-regional connectivity. According to the model, activity in higher regions such as FEF fluctuates at a slower temporal scale than activity in early sensory regions such as V1/V2 (*Murray et al., 2014*; *Honey et al., 2012*; *Chaudhuri et al., 2015*). We simulated the effects of local inhibition of early and late visual areas to test whether the proposed temporal gradient from sensory areas (fast) to association areas (slow) matched the observed changes in connectivity following local cortical perturbations in human participants. Our experimental and modelling results suggest that local inhibition of early visual cortex reduces the discrepancy in endogenous synchronization between lower and higher levels of the visual cortical hierarchy, whereas inhibition of FEF increases the discrepancy. Our work provides novel insights into the neural mechanisms that underlie the effects of local inhibition on large-scale brain dynamics.

## Results

Twenty-three healthy adult volunteers participated in two combined TMS-rsfMRI experiments, conducted on separate days. The number of participants was decided based on previous work adopting a similar experimental design (*Cocchi et al., 2015*; *Eldaief et al., 2011*). A schematic representation of the complete experimental protocol is presented in *Figure 1* (details in the Materials and methods). After exclusion of data from two participants due to excessive head motion or medication, we analyzed resting-state data from 21 of the participants (11 female mean age 27.7, S.D. ± 5.2 years) acquired on a 3T Siemens Trio scanner. Immediately following an initial rsfMRI session, continuous (inhibitory) theta-burst TMS (*Huang et al., 2005*) was applied over one of two different cortical regions of interest (ROIs). A repeat rsfMRI scan was acquired from each participant immediately following TMS stimulation. The anatomical locations of the two TMS targets were: (1) Right early visual cortex – the posterior aspect of the medial occipital lobe (MNI centroid x = 25, y = −92, z = −9), corresponding with areas V1/V2; and (2) Right FEF (MNI centroid x = 31 y = −2, z = 47) – a brain hub within the frontal cortex (see Materials and methods for target region identification). The exact locations of the stimulation sites for each participant are shown in *Figure 1—figure supplement 1*. During TMS, participants closed their eyes and were seated to ensure a relaxed muscle state. This procedure ensured that local stimulation was delivered with participants in a state of rest and in the absence of structured visual input. The order of stimulation of the two cortical sites was counterbalanced across participants.

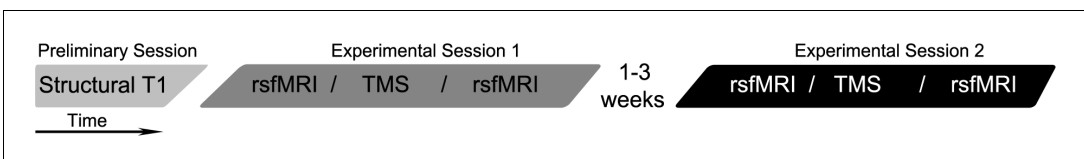

**Figure 1.** Schematic representation of the experimental protocol. Participants initially underwent a high-resolution structural MR scan, which was used to define subject-specific coordinates for subsequent sites of TMS (see Materials and methods for further detail). In Experimental Session 1 and Experimental Session 2, participants undertook 12 min of resting-state functional magnetic resonance imaging (rsfMRI) immediately before and after inhibitory theta-burst stimulation (*Huang et al., 2005*) was applied to one of the two right hemisphere sites (occipital pole or frontal eye fields). All participants underwent stimulation of each of the two target sites across separate sessions. For safety reasons and to avoid carry-over effects of TMS across sessions, Experimental Sessions 1 and 2 were conducted on different days, separated by at least one week, but no more than three weeks (*Huang et al., 2005*). The order of stimulation of the two cortical sites was counterbalanced across participants. Abbreviations: T1= structural scan; TMS = transcranial magnetic stimulation.
The following figure supplement is available for figure 1:

**Figure supplement 1.** TMS target locations.

## Intrinsic functional connectivity of right early visual cortex and frontal eye fields

To characterize the intrinsic connectivity profiles of the two seed regions-of-interest (ROIs; 7.5 mm radius) at baseline, we conducted seed-to-whole brain functional connectivity analyses of the resting-state data acquired *prior to focal stimulation*. These analyses quantified the correlation between the blood-oxygen-level dependent (BOLD) signal timecourse extracted from the seed ROIs and the timecourses extracted from all other brain voxels (see Materials and methods). The ROIs were centered on coordinates within the right occipital pole and the right FEF, as specified above.

As shown in *Figure 2*, activity within the right occipital seed (V1/V2) was positively correlated with activity in other nodes of the visual network, including the lingual gyri, fusiform gyri, the lateral occipital cortex, and the cuneus bilaterally (*Yeo et al., 2011*; *Power et al., 2011*; *Gordon et al., 2016*) ($p < 0.05$ familywise error corrected at cluster level (FWE), *Figure 2a* – red). By contrast, activity in the same occipital seed was negatively correlated (i.e., anticorrelated) with activity in bilateral FEF, the left supramarginal gyrus, the left inferior post-central gyrus, and the left posterior insula

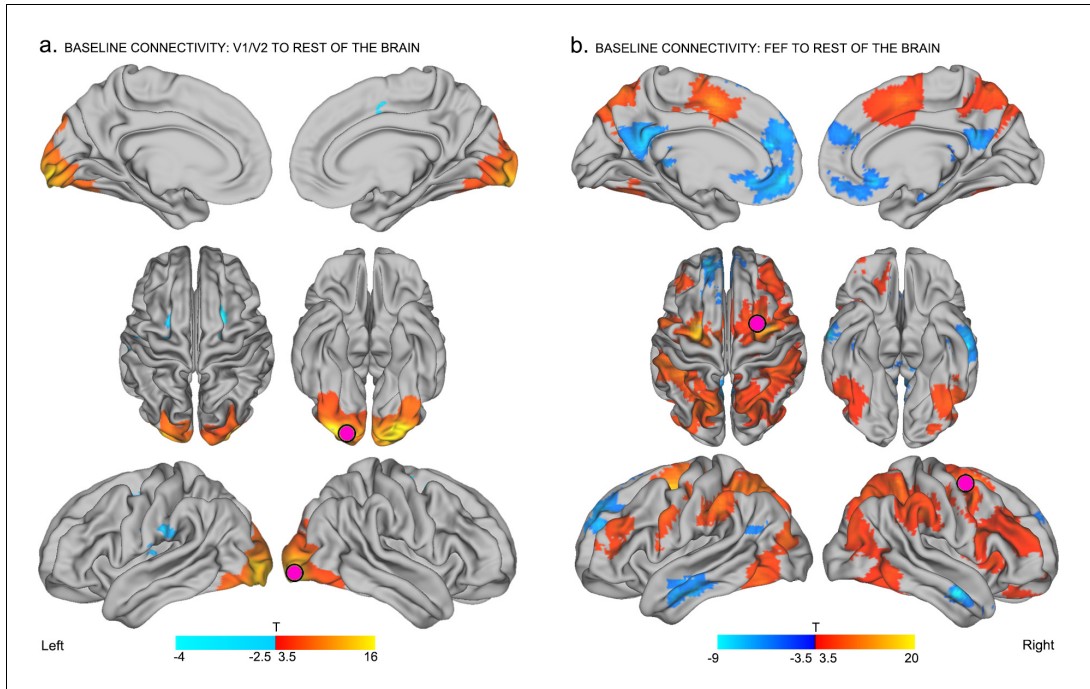

**Figure 2.** Baseline connectivity between TMS-targeted regions and the rest of the brain before stimulation. (**a**) Regions with higher functional connectivity with early visual cortex at the right occipital pole (site of subsequent inhibitory TMS) in the baseline resting-state (i.e., before TMS). At baseline, activity in the frontal eye fields (FEF), left supramarginal gyrus, left inferior postcentral gyrus, and left insula was anticorrelated with activity in right early visual cortex (corresponding to the to-be-targeted region of the occipital pole; pink circle). (**b**) Regions with positive (red – yellow) and negative (blue – light blue) functional connectivity with the right FEF at baseline. The right FEF showed a diffuse pattern of connectivity encompassing frontal, parietal, and temporal cortical areas. Regions known to be part of the default-mode network (medial prefrontal cortex, posterior cingulate, angular gyrus, and medial temporal gyrus) were significantly anticorrelated with the right FEF at baseline. All results are $p < 0.05$ FWE corrected at cluster level.

The following figure supplements are available for figure 2:

**Figure supplement 1.** Effects of seed ROI size on baseline connectivity between TMS-targeted regions and the rest of the brain before stimulation.

**Figure supplement 2.** Control analyses on baseline connectivity between TMS-targeted regions and the rest of the brain.

(*Figure 2a* – blue, p<0.05 FWE). These results confirm and extend previous findings by showing that in the absence of active visual task demands, interactions between early visual cortex and the right FEF can be anticorrelated (*Yeo et al., 2011*; *Power et al., 2011*; *Gordon et al., 2016*; *Ekstrom et al., 2008*).

Neural activity in the right FEF seed was positively correlated with activity in bilateral supplementary motor areas, medial dorsal cingulate, bilateral dorsolateral prefrontal cortex, bilateral superior parietal cortex, and extrastriate visual areas including bilateral precuneus, middle occipital gyri, and inferior temporal gyri (*Figure 2b* – red, p<0.05 FWE). Activity within the right FEF seed was also negatively correlated with activity in default mode regions encompassing the medial prefrontal cortex, posterior cingulate cortex, angular gyrus and bilateral medial temporal gyri (*Power et al., 2011*) (*Figure 2b* – blue, p<0.05 FWE). These findings are consistent with those of previous neuroimaging studies of intrinsic functional connectivity profiles of human FEF (*Yeo et al., 2011*; *Hutchison et al., 2012*), and related findings in macaque (*Vincent et al., 2007*; *Hutchison et al., 2012*). Specifically, our results support the idea that the FEF is a functional brain hub, with widespread connections to a variety of resting-state networks including the fronto-parietal and default-mode systems (*Power et al., 2013*; *Fornito et al., 2016*; *van den Heuvel, 2013a*, *2013b*).

Note that the aforementioned baseline effects were replicated using different sized TMS-seed regions (see *Figure 2—figure supplement 1*), baseline data (*Figure 2—figure supplement 2*) and preprocessing procedures (including global signal regression; see Materials and methods).

## Effects of local inhibitory TMS on functional connectivity

Having established baseline patterns of whole-brain connectivity for the two seed regions at rest, we next examined the influence of local inhibitory TMS on this network activity. To this end, we compared patterns of functional connectivity before and immediately after application of TMS over the right occipital pole and FEF target sites (see Materials and methods).

Inhibitory TMS of right visual cortex (V1/V2) resulted in the emergence of positive correlations between this region and bilateral FEF (*Figure 3—figure supplement 1* and *Supplementary file 2*). TMS of the visual cortex also resulted in the emergence/increase of positive correlations between V1/V2 and extrastriate visual regions including the lingual gyri, the lateral occipital cortex and the parietal cortex (p<0.05 FWE; *Figure 3a* – red, details in *Supplementary files 1* and *2*). These results were replicated when we adjusted the radius of the seed regions to 10 mm and 15 mm, and across different data preprocessing pipelines (*Figure 3—figure supplement 2* and Materials and methods). On the other hand, inhibitory TMS of right FEF resulted in a reduction in positive correlations between the targeted FEF region and visual areas encompassing the bilateral fusiform and occipital gyri (*Figure 3b* – blue, details in *Supplementary files 1* and *2*). Overall, these effects were replicated using TMS-seed regions of 10 mm and 15 mm radius (see *Figure 3—figure supplement 2*) and using different preprocessing procedures (including global signal regression; see Materials and methods). Note that the individual sites of stimulation could not be unequivocally linked with specific effects on functional connectivity (*Figure 3—figure supplement 3*).

We also assessed changes in functional connectivity pre- and post-TMS (for both V1/V2 and FEF sessions) using a control region ipsilateral to the TMS stimulation site but outside the networks of interest (i.e., the inferior portion of the right motor cortex, x = 51, y = −10, z = 18). These analyses revealed no significant changes in functional connectivity between the pre- and post-TMS scans, for both V1/V2 and FEF sessions.

## Impact of variability in the local response to TMS on group-level connectivity

It is well known that the local effects of TMS can vary considerably across participants (*Hamada et al., 2013*). To investigate this issue, we examined whether TMS-induced changes in intrinsic connectivity were related to individual differences in local changes in BOLD signal fluctuations at the site of stimulation. We first extracted the mean amplitude of low-frequency fluctuations in BOLD (ALFF [*Yang et al., 2007*]) from the TMS-target site for each individual participant (see Materials and methods; details in *Supplementary file 3*). We then determined the correlation between TMS-induced changes in local ALFF and the change in functional connectivity between the

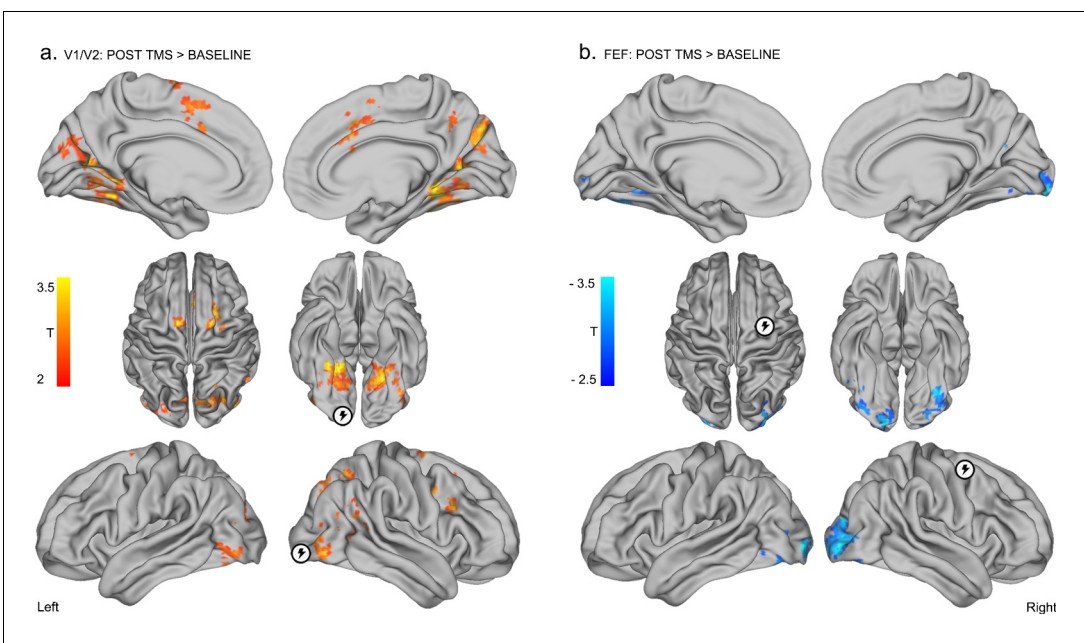

**Figure 3.** Distinct effects of local inhibitory TMS over early visual cortex and FEF. (**a**) Inhibitory TMS of early visual cortex (right occipital pole; encircled lightning symbol) was associated with the emergence of positive correlations between BOLD signals in V1/V2 and bilateral FEF, and the emergence/increase of positive correlations between the V1/V2 seed region and bilateral occipital and parietal cortices (see **Supplementary file 1** and **2** for details, p<0.05 FWE corrected at cluster level). (**b**) Inhibitory TMS of the right FEF (encircled lightning symbol) resulted in the reduction of positive correlations between this target region and bilateral occipital visual areas (p<0.05 FWE corrected at cluster level).

The following figure supplements are available for figure 3:

**Figure supplement 1.** Overlap of baseline and post-TMS connectivity between the early visual cortex seed and FEF, bilaterally.

**Figure supplement 2.** Impact of seed ROI size on effects of local inhibitory TMS over early visual cortex (V1/V2) and FEF.

**Figure supplement 3.** Relationship between TMS sites and changes in brain activity.

**Figure supplement 4.** Relationship between local TMS-induced changes in neural activity and widespread modulation of functional connectivity.

**Figure supplement 5.** Dynamic Causal Modelling used to determine the direction of changes in functional connectivity following TMS of V1/V2 and FEF.

---

targeted ROIs for each participant, with the significant clusters identified at the group level (**Figure 3**).

There was a significant correlation between TMS-induced modulation of the amplitude of slow signal fluctuations (ALFF) and TMS-induced changes in functional connectivity for the right FEF (right FEF ROI to left visual cluster r = 0.6, p<0.01; right FEF ROI to right visual cluster r = 0.6, p<0.01) and for the right occipital pole (right occipital ROI to left visual cluster r = 0.4, trend-level p=0.05) (**Figure 3—figure supplement 4**). This analysis suggests that local TMS-induced reductions in neural activity, as indexed by the amplitude of slow fluctuations in the local BOLD signal, were associated with changes in functional connectivity across visual areas. Once again, these significant correlations were robust across different preprocessing procedures, including deletion (scrubbing) of motion-contaminated volumes (**Power et al., 2012**). Moreover, the effects remained after controlling for

changes in mean BOLD signal between baseline and post-TMS sessions (FEF target seed to left visual cluster r = 0.5, p=0.01; FEF target seed to right visual cluster r = 0.5, p=0.01; occipital target seed to left visual cluster r = 0.4, p=0.05). Importantly, there was no significant relationship between TMS-induced changes in functional connectivity and ALFF values in the baseline and post-TMS sessions.

As a further control, we also examined activity within homologous visual areas in the opposite (non-stimulated) hemisphere. There were *no* significant correlations between changes in the amplitudes of slow signal fluctuations (ALFF values) extracted from homologous visual areas in the left hemisphere (i.e., contralateral to the TMS-target sites in the right hemisphere) and changes in connectivity (left FEF seed to left visual cluster r = 0.2, p=0.42; left FEF seed to right visual cluster r = 0.1, p=0.71; left occipital seed to left visual cluster r = 0.2, p=0.12; left occipital seed to right visual cluster r = 0.2, p=0.51). Finally, there were no clear associations between changes in ALFF and individual TMS target sites (*Figure 3—figure supplement 3*).

## Stochastic dynamic causal modelling (DCM)

To further explore the nature and directionality of TMS-induced perturbations in functional connectivity, we employed stochastic dynamic causal modelling (DCM) (*Daunizeau et al., 2012*). DCM contributes to the analysis by modelling how one region exerts influence over another (i.e., effective connectivity, [*Friston and Harrison, 2003*]), which is not possible using traditional functional connectivity analysis. To this end, we generated two models comprising bidirectional links between the V1/V2 or FEF seed regions and the clusters showing significant changes in connectivity following TMS (*Figure 3*, details in the Materials and methods). DCM analyses of model parameters (pre- versus post-TMS) suggested that stimulation of V1/V2 altered the feedforward influence of V1/V2 on the right occipito-parietal cortex (details in *Figure 3—figure supplement 5* and *Supplementary file 1*, paired t-test p=0.04). Conversely, stimulation of FEF significantly decreased the feedback influence of this region upon V1/V2 (*Figure 3—figure supplement 5*, p=0.0026 for the right cluster and p=0.0012 for the left cluster). Thus, for both V1/V2 and FEF, modulation of effective connectivity induced by focal TMS spread from the stimulation site to distant cortical regions.

## Computational modelling

Results from functional and effective connectivity analyses suggest that V1/V2 stimulation (*Figure 3*, *Figure 3—figure supplement 5*) *increased* feedforward interactions between this cortical area and higher visual cortical areas. Conversely, reduced positive correlations following FEF stimulation were driven by a *reduction* in feedback connectivity (i.e., a reduced influence of FEF on V1/V2). Based on evidence from empirical and computational work (*Hasson et al., 2008*; *Murray et al., 2014*; *Lerner et al., 2011*; *Honey et al., 2012*; *Gauthier et al., 2012*; *Gollo et al., 2015*), we hypothesized that these opposing effects of inhibitory TMS on widespread connectivity might be accounted for by the different timescales at which sensory brain regions and network hubs fluctuate in their levels of activity. To test this hypothesis, we used the Kuramoto model of coupled heterogeneous oscillators, constrained by knowledge of whole brain anatomical connectivity and then adjusted to maximally match the acquired resting-state data at baseline (see Materials and methods for details). The effect of inhibitory TMS was simulated by slowing down (by a variable amount) the intrinsic frequency of the target region. This decision was motivated by the observation that changes in connectivity induced by inhibitory TMS were related to a reduction in the power of the local BOLD signal. Such an energy decrease is embodied in our oscillatory model by a slowdown in the intrinsic oscillatory frequency. The model simulates the dynamics of the whole brain at a relatively high resolution (513 uniformly sized cortical and sub-cortical regions). We focused on the differences in functional connectivity between the two regions targeted by TMS (V1/V2 and FEF) and the rest of the brain. Simulation results showed that virtual inhibition of right V1/V2 within the model increased the positive correlations between this region and the rest of the brain (red in *Figure 4a*), consistent with the group effects we observed following actual TMS over this region in our participant group. This increase in connectivity was robust across a broad range of parameters (i.e., slowing of intrinsic frequencies, omega, *Figure 4a*). Importantly, we found that the simulated effect of local V1/V2 inhibition was due mainly to a significant *increase* in connectivity between this node and other nodes encompassing occipito-temporal and frontal areas of the right hemisphere (*Figure 4b*). Conversely,

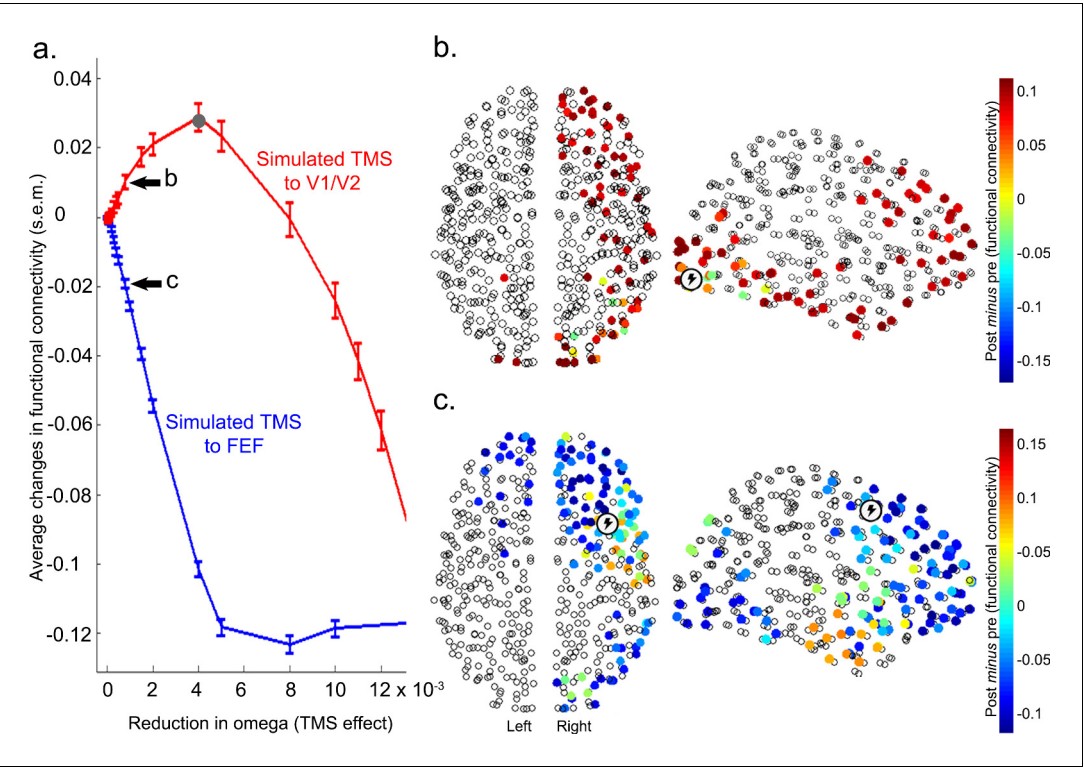

**Figure 4.** Modelling the effects of local inhibitory TMS on diffuse network connectivity. (**a**) In line with the experimental results, computational modelling showed that inhibition of the intrinsic frequencies of V1/V2 and FEF (see Materials and methods) had opposite effects on how these regions were connected with the rest of the brain. Specifically, increasing inhibition of V1/V2 enhanced functional connectivity (i.e., positive correlations) with the rest of the brain until the natural frequency of V1/V2 matched the mean frequency of the whole brain (gray dot). Conversely, inhibition of FEF reduced its positive correlations, or enhanced its anticorrelations, with other brain regions (see also ***Figure 4—figure supplement 1***). s.e.m.= standard error of the mean. (**b**) Following a reduction in its natural frequency, V1/V2 significantly increased its connectivity with other cortical regions comprising occipito-temporal and frontal areas (including FEF; Wilcoxon signed-rank test Bonferroni corrected for multiple comparisons, $p < 2 \times 10^{-6}$). (**c**) A significant reduction in correlations and increase in anticorrelations between FEF and surrounding frontal areas, as well as visual occipito-temporal areas, was observed after slowing of the natural frequency of FEF (Wilcoxon signed-rank test, $p < 2 \times 10^{-6}$). The reduction in omega for panels (**b**) and (**c**) was 0.0008.

The following figure supplements are available for figure 4:

**Figure supplement 1.** Effects of virtual inhibitory TMS on simulated functional connectivity between FEF and V1/V2.

**Figure supplement 2.** Effects of simulated TMS over two control regions - right postcentral gyrus (part of the sensorimotor network) and left superior temporal gyrus (part of the auditory network) – on functional connectivity.

**Figure supplement 3.** Baseline distribution of natural frequencies (omega) in the 513 nodes.

**Figure supplement 4.** Effects of local inhibition on actual and simulated functional connectivity between the target region (V1/V2 or FEF) and the rest of the brain.

simulated inhibition of FEF, again by slowing its intrinsic frequency, was associated with the reduction of correlations in this region's connectivity with the rest of the brain (blue in ***Figure 4a*** and ***Figure 4—figure supplement 1***). This significant effect involved right frontal areas surrounding FEF, and occipito-temporal cortices (***Figure 4c***). Overall, the effects observed within the model were

consistent with the effects observed following actual TMS of V1/V2 and FEF in the experimental participants.

To test whether the opposing changes in functional connectivity revealed by the simulations were driven by non-specific effects of local stimulation, we performed several additional simulations. First, we assessed whether inhibition (delta ω of 0.001) of cortical regions not relevant for our hypotheses – in this instance areas in the vicinity of the right postcentral gyrus and left superior temporal gyrus – could also produce effects on functional connectivity similar to those observed after simulated inhibition of V1/V2 or FEF. Second, we tested whether inhibition of these two control regions would cause specific changes in functional connectivity between early visual cortex and FEF. Simulated inhibition of either control region yielded a markedly different change in connectivity than that observed following simulated inhibition of V1/V2 or FEF (see *Figure 4—figure supplement 2*). Moreover, connectivity between V1/V2 and FEF was not affected by simulated inhibition of either of the two control regions.

## Discussion

We examined dynamic reconfigurations of intrinsic interactions between cortical areas at lower and higher ends of the visual hierarchy in humans following local stimulation with TMS. We applied inhibitory theta-burst stimulation via TMS to one of two regions – early visual cortex and FEF – within the right hemisphere, and examined changes in functional and effective connectivity in the resting-state. Local inhibitory TMS of these regions generated opposing patterns of connectivity between early and late areas within the cortical hierarchy. These distinct effects are consistent with an intrinsic time-scale-based cortical hierarchy (*Murray et al., 2014*; *Bassett et al., 2013*; *Lerner et al., 2011*; *Honey et al., 2012*; *Gauthier et al., 2012*; *Gollo et al., 2015*; *Chaudhuri et al., 2015*), with early visual areas (V1/V2) exhibiting shorter timescales than those of higher areas (such as FEF). By incorporating these timescale effects, the computational model provides supporting evidence for opposing changes in diffuse connectivity following focal inhibitory TMS in the human brain. This finding highlights that inter-regional effects of focal TMS can be predicted by the hierarchical organization of timescales across the cerebral cortex.

At baseline, prior to inhibitory stimulation, neural activity in early visual cortex was *anticorrelated* with activity in the FEF bilaterally. Previous neuroimaging work in humans and monkey has suggested little or no functional coupling between FEF and V1/V2 in the resting state (*Vincent et al., 2007*; *Belcher et al., 2013*; *Damoiseaux et al., 2006*; *Yeo et al., 2011*; *Power et al., 2011*; *Mantini et al., 2013*; *Gordon et al., 2016*). For instance, areas V1 and V2 lie within a primary visual network that does not include FEF (*Yeo et al., 2011*). Our results support previous work by showing a functional segregation between early and late areas within the visual cortical hierarchy at rest. This finding is consistent with the notion that task-induced changes in sensory visual areas may drive the emergence of functional integration between levels of the visual cortical hierarchy (*Ekstrom et al., 2008*; *Roelfsema, 2006*).

A novel finding of our study is that inhibition of early visual cortex resulted in the emergence of positive correlations between this region and FEF, along with other extrastriate and parietal visual areas. Previous studies have investigated intrinsic reconfigurations of large-scale neural systems following focal perturbations of peripheral sensorimotor regions (e.g., *Cocchi et al., 2015*; *O'Shea et al., 2007*; *Grefkes and Fink, 2011*). In line with our results, fMRI studies have shown that local perturbation of primary motor areas can lead to recruitment of related areas (*O'Shea et al., 2007*; *Grefkes and Fink, 2011*; *Cocchi et al., 2015*). For example, inhibitory TMS of the primary motor cortex in healthy adults has recently been shown to increase the strength of resting-state functional connectivity within the sensorimotor system (*Cocchi et al., 2015*). Our finding that unilateral inhibition of early visual cortex increases the positive coupling with FEF and other visual areas adds to a growing literature suggesting that local perturbations of neural activity in peripheral nodes recruit functionally related brain regions (*Cocchi et al., 2015*; *O'Shea et al., 2007*; *Grefkes and Fink, 2011*; *Bestmann et al., 2010*). Specifically, our computational modelling suggests that the effect of inhibitory TMS of early visual cortex might be due to slowing of intrinsic timescales within V1/V2, so that they become closer to the slow intrinsic fluctuations of higher cortical regions such as the parietal cortex and FEF (*Murray et al., 2014*; *Honey et al., 2012*; *Gollo et al., 2015*; *Chaudhuri et al., 2015*).

In contrast to the increased positive coupling apparent after inhibition of early visual cortex, an identical inhibitory TMS protocol delivered over the right FEF led to decoupling between the target region and bilateral early visual cortex. Results from the DCM analysis suggest that such functional decoupling was related to a significant decrease in feedback signals from FEF to early visual cortex. In keeping with the neural mechanism proposed to explain increased connectivity following the inhibition of V1/V2, results from our computational model suggest that the observed reduction in feedback modulation from FEF to V1/V2 can be explained by a further slowing of local FEF dynamics by TMS. The observed changes in connectivity following inhibitory TMS may therefore be explained in terms of a reduction in synchronization discrepancy following local stimulation of V1/V2 and an increase in synchronization discrepancy following stimulation of FEF. Within this broad context, preliminary analyses suggest that there are a host of topographical specificities and nuances that accompany virtual simulations. Further work is needed to address the specificity of our model and the source of possible discrepancies between virtual and empirical findings.

Our findings also reveal a relationship between the magnitude of local signal change and remote modulations in intrinsic functional connectivity. We found that TMS-induced changes in local BOLD signal amplitude at the stimulation site were related to remote modulations in functional connectivity. This suggests that variations in ongoing low-frequency (<0.1 Hz) fluctuations may be a marker of TMS-induced modulation of widespread cortical connectivity.

The dynamic integration of information between sensory and association regions of the cortex is essential for normal brain function. Here we combined functional brain imaging, neural stimulation and computational modelling to elucidate the neural mechanisms that support the emergence and dissolution of interactions between cortical regions within the human visual system following local perturbations in neural activity. Our results suggest that the selective effect of local inhibitory TMS on diffuse patterns of connectivity can be accounted for by an intrinsic hierarchical ordering of cortical timescales (*Murray et al., 2014*; *Honey et al., 2012*; *Gollo et al., 2015*).

## Materials and methods

The study was approved by The University of Queensland Human Research Ethics Committee. Written informed consent was obtained for all participants.

### TMS parameters

Target regions for inhibitory theta-burst TMS were defined using high-resolution structural T1 3D images obtained for each participant and loaded into an ANT Visor Neuro-navigation system with NDI Polaris Spectra infrared camera. The two TMS target sites were identified prior to the first experimental session (*Figure 1*). The anatomical locations of the TMS target coordinates were manually refined according to each individual participant's anatomy. Specifically, if the target region was located in a sulcus, the location for TMS was moved to the gyrus closest to the centroid coordinate (V1/V2: MNI centroid x = 25, y = −92, z = −9; FEF: MNI centroid x = 31 y = −2, z = 47). The early visual cortex target was located anatomically within the occipital pole, posterior to the descending occipital gyrus laterally and the lingual gyrus medially (corresponding to areas V1/V2 [*Thiebaut de Schotten et al., 2014*]). In line with previous studies, the FEF target was located anatomically within the posterior middle frontal gyrus, immediately ventral to the junction of the superior frontal sulcus and ascending limb of the pre-central sulcus (*Ruff et al., 2006*; *Heinen et al., 2014*).

A continuous theta-burst TMS protocol was utilized to induce local inhibition of cortical activity using a previously validated protocol (*Huang et al., 2005*). The inhibitory TMS protocol involved uninterrupted (40 seconds) bursts of 3 TMS pulses delivered at 50 Hz, repeated at 200 ms intervals. TMS was administered using a figure-of-eight coil (70 mm diameter). For FEF stimulation, the TMS coil handle was held at a 45-degree angle to the sagittal plane (*Nyffeler et al., 2006*). For stimulation of the right occipital pole, the coil was oriented with the handle pointing to the right (*Kammer et al., 2001*).

### Establishment of TMS intensity

The intensity of the inhibitory stimulation of the two cortical ROIs was set to 80% of the active motor threshold (*Huang et al., 2005*; *Cocchi et al., 2015*). The active motor threshold was defined as the minimum TMS intensity required to trigger a motor evoked potential (MEP) > 200 μV in at least

three out of five consecutive trials while participants were actively contracting their hand muscle (using a pincer grip) at a level equivalent to ~20% of their maximum voluntary contraction (*Huang et al., 2005*). The location used to establish the active motor threshold was identified with single-pulses of TMS over the right hemisphere. The TMS coil was systematically moved until the optimal cortical site to induce the largest and most reliable motor response (motor evoked potential, MEP) in a muscle of the left hand (the *abductor pollicis brevis (APB) muscle*) was established. MEPs were recorded using surface electromyography (EMG) electrodes (Ag-AgCl) from the left APB. Electromyography signals were amplified (x1000) and filtered (5–500 Hz) using a Neurolog system (Digitimer, UK), and digitised (20 kHz) using a data acquisition interface (BNC-2110; National Instruments) and custom Matlab software (MathWorks [Natick, Massachusetts], see *Source code 1*).

## Resting-state fMRI data acquisition

During resting state fMRI scans, participants were instructed to keep their eyes open and to fixate on a central white cross on a uniform black background. Participants were instructed to let their minds wander freely during the scan. Eye tracking video software was employed to ensure that participants kept their eyes open and looked straight ahead throughout the sessions of resting-state fMRI data acquisition. The eye-open resting-state protocol was preferred over the eye-closed protocol due to recent concerns about controlling wakefulness when participants have their eyes closed (*Tagliazucchi and Laufs, 2014*).

Neuroimaging data were acquired using a 3T Siemens Trio scanner equipped with a 32-channel head coil at The University of Queensland's Centre for Advanced Imaging (Australia). Whole brain T2* images were acquired using an echo-planar imaging sequence (38 axial slices, 320 volumes, gap = 10%, slice thickness = 3 mm, in-plane resolution = 64 × 64, time repetition = 2250 ms, time echo = 28 ms, flip angle = 90°, FOV= 210 × 210 mm, descending slice acquisition). T1 3D images were acquired using the following parameters: 192 axial slices, slice thickness = 0.9 mm; in-plane resolution = 64 × 64, time repetition = 1900 ms, flip angle= 9°, time echo = 2.32 ms, FOV = 230 × 230 mm.

## Preprocessing of neuroimaging data

Preprocessing of the resting-state fMRI data was performed using the Data Processing Assistant for Resting-State fMRI 3 (*Chao-Gan and Yu-Feng, 2010*). The first 10 image volumes were discarded to allow tissue magnetization to reach a steady-state and to provide participants with an opportunity to adapt to the MR scanner environment. DICOM images were converted to Nifti format and underwent slice time correction. To improve normalization, individual participant structural images (T1) were coregistered to functional images using the DARTEL algorithm implemented within the Matlab toolbox SPM8. Tissue segmentation (gray matter, white matter and cerebrospinal fluid) was performed to improve the characterization of non-neural signals in subject space. The following nuisance covariates were regressed from each voxel's time series: six head motion parameters, linear trends, volume-level mean of frame-to-frame displacements greater than 0.4 mm (including the preceding and two subsequent frames [*Power et al., 2014*]) and signals related to time-series unlikely to be modulated by neural activity (CompCor method [*Behzadi et al., 2007*]). After covariate regression, images were normalized to standard MNI space, smoothed using a Gaussian function with a 6 mm full width at half maximum kernel. Finally, a temporal bandpass filter was applied retaining frequencies between 0.01–0.1 Hz.

Supplementary analyses demonstrated that both the mean frame-wise displacement (*Power et al., 2014*) and the number of scrubbed volumes were not significantly different for the pre- and post-TMS scans, in either experimental session. Note that after scrubbing motion-contaminated volumes (*Power et al., 2014*), the number of remaining volumes for each functional scanning session exceeded 8.8 min, which is sufficient time to capture stable correlation coefficients (*Van Dijk et al., 2010*). Thus, the results reported reflect the outcomes following motion correction (i.e., realignment, regression of the six head motion parameters and scrubbing). Given recent concerns in the literature regarding the use of global signal regression, we did not regress the mean global signal during pre-processing. The mean global signal was not significantly different across pre- and post-TMS sessions, for either stimulation site (occipital TMS session: $t_{20}$ = 1.22, p=0.23, FEF TMS session: $t_{20}$ = 0.941, p=0.35). Nevertheless, to ensure that this methodological decision did not alter

the main results reported here, we re-ran the analysis incorporating regression of the mean BOLD signal, and obtained similar results.

## Neuroimaging analysis

### Seed-to-voxel analysis

Seed-to-voxel correlation analyses were conducted to examine patterns of functional connectivity between seed ROIs (TMS target sites) with voxels in the rest of the brain. We first extracted the mean timeseries across grey matter voxels within a quasi-spherical grey matter seed (radius ~7.5 mm) centered on the group-level coordinates in the right occipital pole (MNI centroid x = 25, y = −92, z = −9) and right FEF (MNI centroid x = 31 y = −2, z = 47). Note that all of the results presented in the main text were obtained using the grey matter ROIs with a radius of 7.5 mm. While this ROI dimension was found to best encompass the participants' targeted brain areas in standard MNI space, the main findings were also replicated using additional ROIs with radii of 10 mm and 15 mm; see *Figure 2—figure supplement 1* and *Figure 3—figure supplement 2*. We then conducted a seed-to-voxel analysis, whereby the mean timeseries extracted within each seed ROI was correlated with the timeseries extracted from each voxel in the rest of the brain. This analysis enabled the generation of subject-level spatial maps representing correlation coefficients between the targeted seed with each individual brain voxel for both pre- and post-TMS sessions. The resulting correlation coefficients were Z-transformed to improve the application of univariate second-level statistics.

Seed-to-voxel functional connectivity was calculated at baseline (session specific) using *one-sample* t-tests, for both experimental sessions (V1/V2 and FEF, FWE corrected threshold of p<0.05 at the cluster level, *Figure 2*). Seed-to-voxel functional connectivity was compared across the pre (session specific)- and post-TMS scans using *paired-sample t-tests*, for both resting-state fMRI sessions (V1/V2 and FEF, FWE corrected threshold of p<0.05 at the cluster level, *Figure 3*).

### Analysis of the amplitude of low frequency fluctuations

The amplitude of low frequency fluctuations (*Yang et al., 2007*) (ALFF, 0.01–0.1 Hz) was extracted from each ROI for each individual participant, for both the pre- and post-TMS scanning sessions. We then correlated the TMS-induced change in ALFF with the TMS-induced changes in functional connectivity between the target seed and the significant group level clusters identified in the previous analysis (see *Neuroimaging analysis: Seed-to-voxel analysis)* using bivariate Pearson's correlations.

### Stochastic dynamic causal modelling (DCM)

Effects of TMS stimulation on effective connectivity were analyzed using stochastic DCM implemented in DCM12 (*Daunizeau et al., 2012*). DCM is a computational framework for inferring effective connectivity between cortical regions. Model estimation rests on the Bayesian inversion of state space models by combining dynamic models of neural states with biophysical models of hemodynamics (*Friston et al., 2014*). Importantly, stochastic DCM allows inverting models with uncertainty about the temporal specification of inputs. Because these inputs can represent intrinsic fluctuations, stochastic DCM is particularly useful for model inversion of resting-state data (*Stephan et al., 2010*). For each participant, timeseries were extracted from regions of interest defined by the seed-to-voxel analysis of functional connectivity (*Figure 3*). For both stimulation sessions (*Figure 1*) we used a 7.5 mm spherical seed on targeted regions and main clusters showing a significant TMS effect (*Figure 3*, *Figure 3—figure supplement 5*). This resulted in two distinct models (one for each session) with bidirectional connections between the respective TMS target region (V1/V2 or FEF) and the selected clusters. The models were subsequently inverted to estimate subject-specific parameters. At the group level, the model parameters between the pre- and post-TMS brain states were compared using pairwise two-tailed t-statistics.

## Computational modelling

### The Kuramoto model

We utilized a minimal and standard computational model of synchronization (*Kuramoto, 1984*). In line with previous work (*Hellyer et al., 2015*), we chose this model because it captures the same essential aspects of macroscopic dynamics as more complex models (*Bhowmik and Shanahan,*

*2013*). The Kuramoto model thus provides a good trade-off between complexity and plausibility, incorporating key anatomical constraints and modelling functional elements (*Hellyer et al., 2015*). This model is particularly suitable to assess signal correlations between distinct network nodes and, importantly, is capable of reproducing our experimental results. The model yields a simulation of whole-brain dynamics in which the (consistent) local effect of inhibitory TMS can be represented by reducing a single parameter, as described in detail below. Hence, our simulations allowed us to test the hypothesis that the distinct effects of early- versus late-visual TMS on widespread functional connectivity rely on an intrinsic timescale of neural processes that has been suggested to represent a general organizing principle for the primate cerebral cortex (*Murray et al., 2014*; *Gollo et al., 2015*).

We modelled slow (0.01–0.1 Hz) fluctuations of the BOLD signal in *513* volumetrically similar brain regions (*Zalesky et al., 2010*) as oscillators. Fast (*Breakspear et al., 2010*) and slow (*Schmidt et al., 2015*) cortical oscillations can be modelled with systems of coupled phase oscillators as encompassed within the Kuramoto model. Accordingly, the dynamics of the phase $\theta_i(t)$ corresponding to the oscillator of each region $i$ is given by (*Kuramoto, 1984*),

$$\frac{d\theta_i}{dt} = \omega_i + \lambda \sum_{j=1}^{513} W_{ji} \sin(\theta_j - \theta_i) \qquad (1)$$

where $\omega_i$ is the intrinsic frequency, and $W_{ji}$ is the normalized weight of the anatomical connection between regions $j$ and $i$ extracted from a group-averaged whole-brain structural connectivity matrix $W$ (*Roberts et al., 2016*).

## Gradient of intrinsic timescales

The distribution of $\omega$ is a function of the anatomical node strength $s_i = \sum_{j=1}^{513} W_{ji}$ and is given by

$$\omega_i = b - (b - a)\left(\frac{s_i - s_a}{s_b - s_a}\right)^2 \qquad (2)$$

where $a$= 0.01 Hz, $b$= 0.1 Hz, $s_a$ = min($s$), $s_b$ = max($s$). This distribution is depicted in *Figure 4—figure supplement 3* and acts as a proxy for the hierarchy of timescales, which recapitulates the anatomical hierarchy (*Murray et al., 2014*; *Gollo et al., 2015*). In particular, $\omega_{FEF}$ = 0.0782, $\omega_{V1/V2}$ = 0.0925, and the whole brain mean = 0.0883 were chosen to approximate the frequency of slow resting-state BOLD fluctuations. The parameters $a$ and $b$ were chosen to capture the outer extremes of the BOLD signal frequency bandwidth. The exponent (=2) was tuned to best match the baseline rsfMRI results.

## Anatomical connectivity

The representative connectivity matrix $W$ used in the Kuramoto model – *Equation (1)* – was obtained from a sample of 75 healthy adult participants (*Roberts et al., 2016*). Probabilistic tractography was used to estimate the average number of white matter tracks connecting all pairs of brain regions (513 x 512= 262,656 connections; details in Roberts et al. [*Roberts et al., 2016*]). The resulting structural connectivity matrix was normalized. Probabilistic tractography combined with the relatively large sample size allowed an estimation of the average connectivity strength of pairwise regional connections in the adult population. To ensure that the findings of our model were robust to possible inaccuracies in streamline reconstruction, however, we ran the model using (i) a sparser structural connectivity matrix (30% density vs. original densely connected matrix), and (ii) a correction for the reconstruction of long-range connections (division of the original weights by the fiber distance between any two regions). These analyses replicated the original result showing that TMS of FEF increases the discrepancy between activity in this brain region and V1/V2, whereas TMS of V1/V2 decreases the discrepancy. This robustness is largely explained by the fact that FEF consistently has a high degree (i.e., is a hub) whereas V1/V2 is consistently a peripheral node with a comparatively low degree.

## Model calibration and simulation details

We started by calibrating the model (varying $\lambda$) to closely match the baseline functional connectivity of the two stimulated regions (i.e., the network state prior to application of TMS) with the rest of the brain using the 513-region brain parcellation. To this end, we calculated single-subject 513 x 513 connectivity matrices using the previously preprocessed data (see Preprocessing of neuroimaging data section for details). We then performed 500 simulations for a period of 8 min (discarding 2.12 min of transient time) using the aforementioned model with random initial conditions for varying $\lambda$ values. We then selected and fixed $\lambda$ (=0.0028) for consistency, and verified that small variations did not unduly affect the dynamics. The simulated trials that best replicated the functional connectivity in the real data were selected for each participant. The final correlation between data and simulations for the average functional connectivity between V1/V2 (FEF) and the rest of the brain was r = 0.55 (0.45).

The intrinsic dynamics of each node in the Kuramoto model were reduced to a single parameter, the *natural frequency*. To mimic the putative effect of inhibitory TMS on local oscillations (*Wozniak-Kwasniewska et al., 2014*) we ran numerous simulations (500 trials) with reductions in $\omega$ spanning from $10^{-5}$ to $10^{-2}$ Hz (for both V1/V2 and FEF). These simulations reproduced the topology of changes in functional connectivity between the two stimulation sites – V1/V2 (*Figure 4—figure supplement 4a*) and FEF (*Figure 4—figure supplement 4b*) – and the rest of the brain that were obtained following application of TMS in the participant group. We then quantified the average change in functional connectivity between these regions and the rest of the brain, and identified regions in which the changes in functional connectivity were statistically significant (Wilcoxon signed-rank test, Bonferroni corrected for multiple comparisons).

# Additional information

### Funding

| Funder | Grant reference number | Author |
|---|---|---|
| National Health and Medical Research Council | APP1099082 | Luca Cocchi<br>Martin V Sale<br>Jason B Mattingley<br>Andrew Zalesky |
| ECR grant from the University of Queensland | ECR125 | Luca Cocchi |
| National Health and Medical Research Council | APP1028210 | Martin V Sale |
| National Health and Medical Research Council | APP1110975 | Leonardo L Gollo |
| National Health and Medical Research Council | APP1047648 | Andrew Zalesky |
| Australian Research Council | CE140100007 | Michael Breakspear<br>Jason B Mattingley |
| Australian Research Council | FL110100103 | Jason B Mattingley |

The funders had no role in study design, data collection and interpretation, or the decision to submit the work for publication.

### Author contributions

LC, Conception and design, Acquisition of data, Analysis and interpretation of data, Drafting or revising the article; MVS, Conception and design, Acquisition of data, Drafting or revising the article; LLG, Conceived (with LC and MB) and designed the simulations and performed the simulations and analyzed results, Drafting or revising the article; PTB, VTN, AZ, Analysis and interpretation of data, Drafting or revising the article; MB, Conception and design, Analysis and interpretation of data, Drafting or revising the article; JBM, Conception and design, Drafting or revising the article

## Author ORCIDs

Luca Cocchi, http://orcid.org/0000-0003-3651-2676
Jason B Mattingley, http://orcid.org/0000-0003-0929-9216

## Ethics

Human subjects: The study was approved by The University of Queensland Human Research Ethics Committee. Written informed consent was obtained for all participants.

## Additional files

### Supplementary files

• Supplementary file 1. Changes in TMS-targets to whole brain connectivity.

• Supplementary file 2. Direction of changes between regions targeted with TMS and whole brain functional connectivity. Group-level average values extracted from key regions showing a significant change in functional connectivity after inhibitory TMS (*Figure 3*). Standard error of the mean (in brackets) was calculated on single-subject connectivity values in each region. Regions consisted of a sphere of 7.5mm of radius centered on the co-ordinates indicated (MNI space).

• Supplementary file 3. ALFF values.

• Source code 1. Matlab code used to process electromyography signals.

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
