## [Decision Letter]

Thank you for submitting your article "A hierarchy of timescales explains distinct effects of local inhibition of primary visual cortex and frontal eye fields" for consideration by *eLife*. Your article has been reviewed by three external peer reviewers, and the evaluation has been overseen by Jody Culham as both the Reviewing Editor and Senior Editor.

The reviewers have discussed the reviews with one another and the Editor has drafted this decision to help you prepare a revised submission.

The following individual involved in review of your submission has agreed to reveal his identity: Michael Arcaro (Peer Reviewer).

Summary of manuscript:

The authors combine fMRI, TMS, and computational modeling to investigate interactions between visual cortical regions. They find that cortical stimulation of posterior visual cortex (V1/V2) leads to an increase in BOLD correlations with FEF, but stimulation of FEF leads to a decrease in BOLD correlations with posterior cortex. The modeling work indicates that these changes are related to the intrinsic timescales of these regions. This is a truly impressive study. The multi-methodology approach and research question is very novel and the results have important implications for understanding the neural processes facilitating interactions along the cortical hierarchy.

Summary of reviews:

While all reviewers of the work were quite positive, they raised a number of concerns that should be addressed in a revision. Although the journal policy is usually only to provide a summary of the main points, given that the reviewers raised a number of important and thoughtful points that may be "lost in translation" by condensation, the full reviews are appended below. The Essential Changes are based on the post-review discussion amongst the reviewers and editor and are outlined below. We also strongly recommend the authors review the full list of suggestions from the reviewers and, at their discretion, determine whether they can improve the manuscript based on the constructive criticism provided. The editor did not think that suggested changes requiring collection of new data were essential, though there are some cases where additional analyses of the extant data could be beneficial in addressing the suggestions.

Essential changes:

1) Most imperatively, two reviewers noted that there is some confusion between the concepts of anticorrelation and connectivity changes that is challenging for the interpretation of the results.

Reviewer #3 states, "Positive and negative correlations are not synonymous with increased and decreased connectivity. The authors summarize their findings as "stimulation of early visual cortex selectively increased feedforward interactions with FEF" and " stimulation of FEF decreased feedback interactions with early visual areas." TMS of FEF resulted in a stronger anti-correlations with V1/V2. An argument could easily be made that connectivity increased in both cases, especially since these areas appear to be moderately anti-correlated at rest (see below)."

Similarly Reviewer #2 points out that the interpretation of connectivity changes depends not solely on changes to correlation values but the magnitude of the r value with respect to no correlation (r=0).

This must be clarified with more careful wording, unpacking of the specific effects (Figure 3) and better explanations/discussion.

2) Two reviewers (#2 and #3) raised questions about the localization of the TMS sites. This must be clarified.

3) Reviewer #1 raised concerns about the accuracy of the structural connectome on which the Kuramoto model is based. The other two reviewers agreed this was a concern so this must be discussed.

4) The Reviewing Editor raised a concern about the asymmetry in connectivity between FEF and V1/V2 at baseline and the other reviewers agreed this should be addressed. Specifically, the data in Figure 2 on baseline resting-state connectivity showing that a V1/V2 seed is negatively correlated with FEF while an FEF seed did not show a negative correlation with V1/V2. One would expect these effects to be roughly symmetric and it would be helpful to clarify why they are not. One possibility is that it's just a thresholding issue (e.g., 2b would show V1/V2 at a slightly more liberal threshold). Another possibility is that the seeds are not exactly in the same location as the sites that are correlated with the opposite seed). Please clarify.

Recommended considerations

5) Two reviewers agreed the manuscript would be strengthened by the inclusion of control sites in the connectivity analysis while a third reviewer disagreed. The most useful additional analysis that the majority agreed would be worth investigating would be to examine the connectivity of the V1/V2 site after FEF stimulation and vice versa. We suggest the authors check this out and see if it adds any value to the paper (and include it if it does or provide a brief summary in the reply letter if it doesn't).

Reviewer #1:

This paper presents a combination of TMS with different analyses of functional/effective connectivity and computational modelling, using a whole-brain formulation based on the Kuramato model. This combination is so far unique and enables the authors to provide evidence for a plausible and quite fundamental principle of cortical organisation, i.e., that cortical areas at different hierarchical levels operate at different time scales. I think this is a strong and innovative paper which elegantly combines complementary techniques. However, from my perspective, a few issues would benefit from clarification or reformulation:

1) I think the presentation of the key findings and the conclusions drawn needs to be improved. In brief, the key conclusion seems to be: inhibitory TMS leads to a local slowing of the timescale (frequency of dominant oscillations) at which an area operates; TMS of V1/V2 decreases and TMS of FEF increases the discrepancy in oscillatory frequencies between these two levels of the visual hierarchy; this explains why the former leads to increased bottom-up connectivity and the latter leads to reduced top-down connectivity. The authors get to this on fourth paragraph in the Discussion, but do not make it as explicit as I think they should. Moreover, it would help if this interpretation and the underlying hypothesis would be pointed out more clearly earlier in the paper, e.g. in the Introduction.

2) To support this conclusion, it would be helpful if not only the correlation between changes in ALFF and changes in connectivity were shown; additionally, it would be important to see the magnitude and direction of the local changes in ALFF, separately at both levels of the hierarchy, that are induced by TMS.

3) One potential concern for the whole-brain Kuramato model is the accuracy of the structural connectome on which the model rests. This connectome was derived from diffusion-weighted imaging data, using probabilistic tractography. It would be helpful if the authors could provide some reassurance that these connectivity estimates are robust and not overly affected by methodological problems of tractography, such as the known bias in reconstructing short- versus long-distance connections.

4) Testing the robustness of the functional connectivity results across a variety of preprocessing strategies for resting state data is a strong aspect of this paper. However, it did not entirely get clear what motion correction procedure the results eventually reported are based on. It would be helpful if this could be clarified.

Reviewer #2:

The aim of the study was to explore long-range connectivity changes (assessed by resting-state fMRI) resulting from a local decrease in excitability induced by continuous theta burst TMS (cTBS). The authors used cTBS to modulate excitability in the V1 and the R-FEF in separate sessions. Inhibition of V1 with cTBS led to an increase in functional connectivity between V1 and R-FEF. Inhibition of R-FEF led to a decrease in functional connectivity between R-FEF and V1. This is an interesting study that looks to combine brain stimulation and neuroimaging with modelling to address a clearly articulated question.

Major Comments:

1) The authors suggest that there is a significant relationship between the amplitude of local BOLD signal within the FEF and changes in connectivity between FEF and V1 (Figure 3—figure supplement 3). Could the connectivity changes therefore be (trivially) explained by merely increased or decreased signal-to-noise in the seed region leading to changes in the connectivity? (If there is greater signal – and therefore greater variability – in the seed region, it is more likely that functional connectivity can be identified between that region and elsewhere).

This is a difficult problem to address, and I admit I do not have an easy solution, but it should be at the least discussed. I admit that I do not understand the sentence "the effects remained after controlling for changes in mean BOLD signal between baseline and post-TMS sessions", which may be aimed at addressing this conflict.

2) The authors present no controls for their study. There are a number of questions raised by this:

a) Can they demonstrate that the fluctuations in connectivity between V1 and FEF, separated in time as these scans were, cannot explain the changes here (i.e. is this just an effect of repeated rs-fMRI scanning, rather than a TMS effect)? It may be possible to get this data from existing datasets – a step towards this would be to study the differences between the two baseline sessions, though this would not account for changes due to differing time spent in the scanner.

b) Are these changes specific to the stimulated regions? If the seeds for the functional connectivity analyses are placed elsewhere in the functional networks, can similar changes be elicited? And similarly for regions outside the networks? The authors have the data to perform these analyses.

c) Are the changes seen specific to stimulation at these sites? If the network was stimulated at a different site, would this result in the same pattern of results (i.e. is this just a reflection of some perturbation to the network as a whole or does it reflect the specific connectivity between these regions). This would require more data to be acquired, but is important for their conclusion that these are specific effects. A step towards this might be to show that stimulation of the FEF does not lead to connectivity changes in the V1 seed and vice versa, which they have the data to demonstrate.

3) One of the major things I struggled with in this study is the authors' interpretation of ante-correlations. To me, the demonstration of a significant, negative functional connection between two regions is not a lack of a functional relationship, a point with which the authors seem to agree. Therefore, a numerical increase in the r value between two regions is not an increase in connectivity if it does not go through 0, but rather is a decrease in (inhibitory) connectivity. If it does go through 0 it is then a reversal of negative to positive functional connectivity and the interpretation is very different.

This is an important point of interpretation, and is currently not clear in the manuscript. If I understand correctly, the authors suggest that inhibitory TMS to V1 led to an increase in connectivity between V1 and FEF. These regions were negatively correlated at baseline (Figure 2). Does this increase in connectivity mean that these regions are now positively correlated or that they are less negatively correlated? In the caption to Figure 3 they state that these are "antecorrelations" but I do not see the data to support that.

Likewise, inhibitory TMS to the right FEF led to a significant decrease in functional connectivity between this region and V1. Does this mean that there is now a significant functional connectivity between these regions, where there was not previously?

Reviewer #3:

The authors combine fMRI, TMS, and computational modeling to investigate interactions between visual cortical regions. They find that cortical stimulation of posterior visual cortex (V1/V2) leads to an increase in BOLD correlations with FEF, but stimulation of FEF leads to a decrease in BOLD correlations with posterior cortex. The modeling work indicates that these changes are related to the intrinsic timescales of these regions. This is a truly impressive study. The multi-methodology approach and research question is very novel and the results have important implications on the neural processes facilitating interactions along the cortical hierarchy. The manuscript is well written. Several aspects of the data that could affect the interpretation of the results need clarification. Specifically, the authors should evaluate potential influences of the underlying functional organization on the observed interactions, and address the differences between the imaging and modeling results. In addition, I have concerns on the specificity of the localization for both the imaging data and TMS. Conceptually, the authors also need to better address the relation of positive/negative BOLD correlations to increased and decreased connectivity.

Main Comments:

Positive and negative correlations are not synonymous with increased and decreased connectivity. The authors summarize their findings as "stimulation of early visual cortex selectively increased feedforward interactions with FEF" and "stimulation of FEF decreased feedback interactions with early visual areas." TMS of FEF resulted in a stronger anti-correlations with V1/V2. An argument could easily be made that connectivity increased in both cases, especially since these areas appear to be moderately anti-correlated at rest (see below).

Are the observed interactions between V1/V2 and FEF generalizable to posterior visual vs. higher-order (frontal) regions? The modeling certainly appears to predict generalizable effects, but I'm not sure the imaging data specifically demonstrate this.

Could the correlation patterns be related to the underlying topographic organization? Anatomical and functional studies in monkeys have shown that FEF and visual cortex is topographically organized with foveal and peripheral V1/V2 connected to differentiable portions of FEF (Schall et al. 1995; also see Babapoor-Farrokhran et al. 2013; Janssens et al. 2013). The particular increase/decrease in BOLD co-fluctuations may reflect where stimulation was applied relative to the functional organization of these regions (specifically retinotopy). This would still be an interesting result, but should be clarified.

Given that TMS to FEF results in decreased BOLD activity in foveal V1, but increased BOLD activity in peripheral of V1 (Ruff et al. 2006; Ruff et al. 2009; Driver et al. 2010), were increases in BOLD correlations observed in peripheral V1/V2 for Post-FEF-TMS? Were any decreases in correlations observed in frontal cortex for POST-V1-TMS? If so, would this be predicted by the model?

How precise was the TMS? Judging by the provided MNI coordinates, it's not clear that the closest gyrus on the surface is still within V1/V2. It's very close to lateral occipital extrastriate areas such as LO1/2. Given the lateral/medial distinction of FEF in the macaque, is it possible that TMS stimulation of FEF targeted a more foveal region than the seed region for the BOLD correlations?

The authors should address sources of variability in the localization of areas. Closer evaluation of the correlation maps could potentially address this. While the pre- and post- TMS correlation maps of FEF heavily overlap in the ipsilateral hemisphere (Figure 3—figure supplement 1), there is some separation with the post-TMS increase located slightly more inferior and lateral. Slices end at z = 40 in this figure though. Does the increase in correlation post TMS extend further inferior? What is the relationship between the observed correlations in FEF and the seed FEF?

The model predicts that TMS to V1/V2 leads to a general increase in connectivity across cortex, and TMS to FEF leads to a general decrease. Can the model speak to potential topographically specific interactions as noted above?

What accounts for the differences between the model predictions (Figure 4) and the observed changes in BOLD correlations (Figure 3) post-TMS? Specifically, the model predicts increases in extrastriate cortex, temporal lobe, and much of the frontal lobe after TMS to V1 while changes in BOLD correlation are minimal in non-FEF frontal cortex, the temporal lobe, and extrastriate cortex (with the exception of a surprisingly specific increase in posterior parahippocampal cortex). There also appear to be increases in BOLD correlations within the parieto-occipital sulcus and retrosplenial cortex that are not clearly predicted in the model. After TMS to FEF, the model predicts large decreases in the frontal lobe and anterior, lateral occipital cortex (possibly near area MT) and increases in anterior temporal cortex, which are not apparent in the imaging data.

Reviewing Editor:

I was puzzled by the data in Figure 2 on baseline resting-state connectivity showing that a V1/V2 seed is negatively correlated with FEF while an FEF seed did not show a negative correlation with V1/V2. One would expect these effects to be roughly symmetric and it would be helpful to clarify why they are not. One possibility is that it's just a thresholding issue (e.g., 2b would show V1/V2 at a slightly more liberal threshold). Another possibility is that the seeds are not exactly in the same location as the sites that are correlated with the opposite seed). Please clarify.

I agree with Reviewer #3 that the folded surfaces make it hard to see some sites (esp. FEF) and this could be resolved with partial cortical inflation or other views (e.g., superior view to show FEF).

[Editors' note: further revisions were requested prior to acceptance, as described below.]

Thank you for resubmitting your work entitled "A hierarchy of timescales explains distinct effects of local inhibition of primary visual cortex and frontal eye fields" for further consideration at *eLife*. Your revised article has been evaluated by Eve Marder (Senior Editor), Jody Culham (Reviewing Editor), and 3 reviewers (including Michael Arcaro, who agreed to reveal his name).

The manuscript has been improved considerably and two of the reviewers are largely satisfied. However, there are some remaining issues that still need to be addressed, as outlined below. Although at *eLife*, we try to avoid putting authors through an endless gauntlet of unnecessary revisions, we also aim to ensure that published manuscripts meet our high standards. In this case, the remaining concerns were substantive enough that we think another round of revisions would greatly benefit the clarity and impact of the manuscript.

Again, since the detailed points made by the reviewers were well articulated and are appended in full after the summary from the Reviewing Editor.

Essential points

1) The manuscript has become clearer by rephrasing in terms of changes to positive and negative correlations (rather than just increases or decreases without respect to the initial sign of the correlation as in the first version). However, both Reviewer #3 and the Reviewing Editor still found it hard to interpret without an additional figure to show the post-TMS connectivity. (not just pre and post-pre changes). See their comments for details (Reviewer 3, Point 1; RE, Point 1).

2) Although based on your reply, we understand your justification for displaying data on the folded cortical surfaces, the problem remains that one of your two key areas (FEF) is hardly visible from lateral views. This can be solved simply by presenting either a superior view or a horizontal slice consistently in all figures (not just a subset as it is currently, e.g., slice on Figure 2, superior view on Figure 3). And you may as well add an inferior view (to show lingual/fusiform cortex).

3) In the post-review consultation, the other reviewers reinforced a suggestion from Reviewer #3 (last line of Reviewer 3, Point 1). Reviewer #1 stated: I think the one issue to resolve is whether the model does or does not predict anti-correlations, that is, the authors should clarify the connection between their description in the main text and the results shown by Figure 4.

Recommended considerations

Reviewer #3 raised two points that the other reviewers (in post-review consultation) were less convinced were essential to resolve. Nevertheless, the Reviewing Editor would encourage you to take this feedback into consideration in case you can use it to strengthen the manuscript.

1) Reviewer #3 (Point 2) still questions the asymmetry of the FEF and V1/V2 correlations,. The Reviewing Editor thinks that if it cannot be fully resolved, it should at least be made more apparent in the main manuscript (by showing the FEF clearly in extant figures and/or moving Figure 3—figure supplement 1 to the main text) and discussed. As is, the main figures don't show this because of the views presented).

2) Further consideration of the earlier results of Ruff and colleagues may be warranted (Reviewer 3, Point 3). The other two reviewers were less convinced of this and noted that the TMS methods by Ruff were quite different and your paper wasn't designed to examine the relationship between connectivity and retinotopy. Nevertheless, Reviewer #3 thought the manuscript could be more compelling on this front (for specifics, see "Added Comments from Reviewer #3 during Post-review discussion"). The Reviewing Editor will leave it as "your call" as to whether you can use Reviewer #3's suggestions to strengthen your manuscript or agree more with the other two reviewers that it is beyond the scope of your aims.

Other points should be considered at the authors' discretion.

Reviewer #1:

I have gone through the authors' response and am happy with the additional analyses and revisions. My only (and slight) reservation is that the introductory sentence ("Probabilistic tractography.… allowed an accurate estimation of the average connectivity strength.…") to the section on "Anatomical connectivity" continues to be a little too optimistic, given that there is an ongoing debate about the pitfalls of tractography (e.g., see Reveley et al. 2015, PNAS). Otherwise, I do not see any major remaining problems and would recommend acceptance of the paper.

Reviewer #2:

The authors have provided a thorough and thoughtful set of answers to my comments and I have no further concerns.

Reviewer #3:

The authors made extensive revisions to the manuscript including several new figures and provided a thorough and thoughtful response to the previous reviews. While the revision addressed several issues, my main concern regarding the interpretation of the correlation changes and their relation to the TMS targets and seed regions remain.

1) The relationship between positive / negative correlations and increased / decreased correlations needs further clarification:

Seventh paragraph of Results section. "Inhibitory TMS of right visual cortex (V1/V2) resulted in the emergence of positive correlations between this region and bilateral FEF." – The difference maps in Figure 3 do not illustrate this and there is no reference to statistical measures. Are the post-TMS, positive correlations significantly different from 0? Or is there just a significant difference between baseline and TMS? It could be very informative to show correlation maps post-TMS for V1/V2 and FEF seeds w/o the baseline subtraction (ala Figure 2).

Results section, subsection “Computational modeling”. "Simulation results showed that virtual inhibition of right V1/V2 within the model increased the positive correlations between this region and the rest of the brain (red in Figure 4)," – I'm still confused about correlations in the simulation. This statement suggests that simulated V1/V2 connectivity with FEF started out positive and increased following V1/V2 inhibition. That is not entirely consistent with the imaging data (i.e., correlations were negative at baseline). Figure 4 does not illustrate which simulated correlations are positive / negative. This figure could be expanded to parallel the imaging data by showing the baseline connectivity and connectivity post-simulated TMS.

Same section. "Conversely, simulated inhibition of FEF, again by slowing its intrinsic frequency, was associated with the emergence of significant anticorrelations in this region's connectivity with the rest of the brain (blue in Figure 4)." – In the previous version, this was described as decreased connectivity, which I assumed was a lack of (any) correlation. Does the modeling result actually predict anticorrelations?

2) The authors refer to a common V1/V2 region (also FEF) across TMS targets, seed ROI and observed correlations. However, asymmetry in the correlations and anatomical variability suggest that this is not necessarily the case. The relationship between all three for V1/V2 and FEF needs further clarification.

The data would be more compelling if the seeds were better matched to the observed correlations. For example, the FEF seed could be adjusted to be in better correspondence with the V1/V2 anticorrelations (or vice versa). Further, why not use the FEF seed as an ROI to evaluate correlations pre- and post- V1 TMS (and vice versa)? An ROI approach would be a more direct way for assessing correlation changes due to TMS (vs. the qualitative assessment of the correlation maps).

The FEF seed is noted as being posterior and lateral to the V1/V2 baseline anticorrelation. From the Figure 1—figure supplement 1, the FEF TMS sites appear to be within the FEF seed, but also extend lateral and posterior, suggesting that the TMS-site for FEF and the anticorrelations with V1/V2 only partially overlap. The authors should better address whether such variability had an affect on the correlations. e.g., the authors could show something similar to Figure 3—figure supplement 3, but color code the FEF TMS sites based on the changes in correlation with V1/V2 (and vice versa). If the precise TMS location did not matter, there should be no difference in the correlations between the subset of TMS sites that overlapped with the V1/V2 baseline anticorrelations and the ones that didn't.

Figure 1—figure supplement 1 nicely illustrates the relationship between the TMS sites and seed area, but could be improved. There is no reason to show the whole brain, and that only makes evaluation of the correspondence more difficult. Focal views of V1/V2 and FEF should be shown (such as in Figure 3—figure supplement 3). This figure would further benefit by illustrating the relation between observed correlation changes and TMS sites. e.g., showing the TMS target locations overlaid on a correlation changes post TMS.

Figure 2—figure supplement 2. The control analysis did not yield significant negative FEF correlations with the V1/V2 seed. Is there still a significant increase in correlation when comparing these correlations to the post-TMS?

3) As stated previously, Ruff and colleagues (2006, 2009) found decreased correlations with foveal V1 and increased correlations with peripheral V1 after TMS to FEF. While their experimental paradigm differed with the current study and the reviewers' comment on state dependent effects is well taken, these prior studies looked at correlations during both task (visual) and rest (non-visual) conditions (and found no significant differences). Such findings are clearly relevant to the current work and should be discussed. How can these prior data (specifically the observation of both increases and decreases in V1 correlations) be reconciled with the current proposal on how TMS differentially affects the timescales of early visual cortex and frontal cortex?

Added Comments from Reviewer #3 during Post-review discussion:

The current study proposes a general difference between early visual and higher order regions with interactions governed by their intrinsic timescales of processing. The Ruff studies found that interactions between early visual cortex and FEF vary depending on the topographic sub-region within early visual cortex. Aside from methodological differences, those data suggest heterogeneity in early visual-frontal interactions (there is also substantial evidence from monkey studies showing heterogeneity of anatomical connections and function within both FEF and V1). It's unclear to me how any such heterogeneity can be directly accounted for by the current study's broad short and long timescale differences in early visual cortex and FEF, respectively. More so, it actually seems that their model predicts the opposite (lack of heterogeneity). I think an argument potentially could be made that any heterogeneity in correlations reflects the heterogeneity in connections, though it's not clear to me what that mapping would be, and their data do currently do not speak to this.

On a related note, higher precision in the targeting could mitigate this issue. I completely agree with [one reviewer] that there is a limit to what the authors can do in regards to the localization. However, they easily could have performed seed-to-seed correlations or used an areal atlas (particularly for V1), which would be much more of a control in localization than what they currently have done. These are very simple analyses, and frankly, I don't understand why these weren't done in the first place.

I completely agree with [another reviewer] that it would not be fair to expect the authors to directly address this in their paper as their model clearly wasn't intended to test retinotopy or any other areal substructure. However, I think it's reasonable to expect the authors to have an idea (discuss) on how their model would fit within the well established architecture of the brain regions they've specifically investigated, particularly since their imaging results look like they do not encompass the whole area of V1 or FEF.

For clarification, I was referring to the eccentricity specific correlations for visual areas V1-V4 from TMSing FEF in the Ruff 2009 paper, which seems to expand upon the V1 result in their 2006 paper.

Reviewing Editor:

In trying to wrap my head around the data given the new presentation in terms of not just the direction of changes but their effect on positive and negative correlations, I had to make visual comparisons of Figure 2 and Figure 3. That is, I was trying to see whether increases/decreases in connectivity (Figure 3) meant increases/decreases in positive/negative correlations. I thought it would be easier for readers to understand this if (1) an additional figure were presented in the main manuscript to show post-TMS connectivity (not just differences in connectivity); and (2) the same brain views were presented in the same order (as is, Figure 1 doesn't show superior views to highlight FEF or inferior views to show the lingual and fusiform gyri).

---

## [Author Response]

Summary of reviews:

*While all reviewers of the work were quite positive, they raised a number of concerns that should be addressed in a revision. Although the journal policy is usually only to provide a summary of the main points, given that the reviewers raised a number of important and thoughtful points that may be "lost in translation" by condensation, the full reviews are appended below. The Essential Changes are based on the post-review discussion amongst the reviewers and editor and are outlined below. We also strongly recommend the authors review the full list of suggestions from the reviewers and, at their discretion, determine whether they can improve the manuscript based on the constructive criticism provided. The editor did not think that suggested changes requiring collection of new data were essential, though there are some cases where additional analyses of the extant data could be beneficial in addressing the suggestions.*

*Essential changes:*

*1) Most imperatively, two reviewers noted that there is some confusion between the concepts of anticorrelation and connectivity changes that is challenging for the interpretation of the results.*

*Reviewer #3 states, "Positive and negative correlations are not synonymous with increased and decreased connectivity. The authors summarize their findings as "stimulation of early visual cortex selectively increased feedforward interactions with FEF" and " stimulation of FEF decreased feedback interactions with early visual areas." TMS of FEF resulted in a stronger anti-correlations with V1/V2. An argument could easily be made that connectivity increased in both cases, especially since these areas appear to be moderately anti-correlated at rest (see below)."*

*Similarly Reviewer #2 points out that the interpretation of connectivity changes depends not solely on changes to correlation values but the magnitude of the r value with respect to no correlation (r=0).*

*This must be clarified with more careful wording, unpacking of the specific effects (Figure 3) and better explanations/discussion.*

We thank the reviewers for highlighting our lack of clarity on this important definition. We agree that the sign of the correlations – positive versus negative – is not synonymous with increased and decreased connectivity, respectively. To avoid any confusion we now present the results as TMS-induced changes in positive or negative correlations (i.e., anticorrelations) relative to baseline.

We have now clarified the text in the section titled *“Effects of local inhibitory TMS on functional connectivity”*:

The following sentences in the Discussion have also been clarified:

Second paragraph:

“A novel finding of our study is that inhibition of early visual cortex resulted in the emergence of positive correlations between this region and FEF, along with other striate and extrastriate areas.”

“Our finding that unilateral inhibition of early visual cortex increases the positive coupling with FEF adds to a growing literature.”

Beginning of the 3^rd^ paragraph:

“In contrast to the increased positive coupling apparent …”

The Results section, subsection “Computational modelling” has also been amended:

“Results from functional and effective connectivity analyses suggest that TMS of V1/V2 stimulation (Figure 3) increased feed forward interactions between this cortical area and higher visual cortical areas. Conversely, enhanced anticorrelations following FEF stimulation were driven by a reduction in feedback connectivity (i.e., a reduced influence of FEF on V1/V2).”

“Simulation results showed that virtual inhibition of right V1/V2 within the model increased the positive correlations between this region and the rest of the brain (red in Figure 4), …”

“Conversely, simulated inhibition of FEF, again by slowing its intrinsic frequency, was associated with the emergence of significant anticorrelations in this region’s connectivity with the rest of the brain (blue in Figure 4).”

The figure legends for Figure 3, Figure 3—figure supplement 3, and Figure 4 have also been amended to be consistent with the revised text. Please also see the revised Figure 3 and Figure 3—figure supplement 2. In these figures we have now used *Post TMS > Baseline* (red= increased positive correlations, blue= increased negative correlations) instead of the ambiguous terms of “increased/decreased connectivity”. Similarly we changed the text in [Supplementary-material SD1-data]. Finally, the section on DCM has also been clarified to highlight the unique contribution of this analysis over correlations between BOLD signals (i.e., functional connectivity).

*2) Two reviewers (#2 and #3) raised questions about the localization of the TMS sites. This must be clarified.*

The localization of stimulation sites was initially based upon MNI coordinates in standard space defining the centroid of the targeted brain regions (e.g., FEF, Power et al., Neuron, 2013). These coordinates were then manually refined using established anatomical landmarks at the level of individual anatomy (e.g., V1/V2, Thiebaut de Schotten M et al., Cortex, 2014). As requested, we now provide a new figure (Figure 1—figure supplement 1) to clarify the location of the stimulation site for each participant.

*3) Reviewer #1 raised concerns about the accuracy of the structural connectome on which the Kuramoto model is based. The other two reviewers agreed this was a concern so this must be discussed.*

The estimation of structural connectivity was performed on data from a large sample of healthy adult participants (N = 75) using state-of-the art probabilistic tractography algorithms. These algorithms have good sensitivity (MRTrix: http://www.mrtrix.org/). However, to test the impact of possible biases in the connectome reconstruction on our computational findings we have now run two further analyses.

First, we re-ran the model using a sparser structural connectivity matrix (i.e., 30% density). Dense structural matrices maximize the sensitivity (i.e., lower false negative in detecting fibers) but have lower specificity (i.e., increased false positives). Conversely, sparser matrices maximize specificity but lack in sensitivity. Running the model using different matrix thresholds changes the balance between sensitivity and specificity. Results from this analysis replicated the main results of the model (i.e., local inhibition of early visual cortex reduces the discrepancy in endogenous synchronization between lower and higher levels of the visual cortical hierarchy, whereas local inhibition of FEF increases the discrepancy).

Second, we re-ran the model using a connectivity matrix that incorporated a correction for the known tractography bias in reconstructing short- versus long-distance connections. Because of the increased likelihood of randomly seeding larger long-range fibers, we have penalized the long-range connections by dividing the weights by the fiber distance (e.g., Hagmann et al., 2008). Model results obtained using this *distance correction* replicated those originally presented. Specifically, the variation in functional connectivity between the original matrix (TMS of FEF -> emergence of anticorrelations with V1/V2 of 0.07; TMS of V1/V2 -> increased positive correlations with FEF of 0.06), the 30% matrix (FEF -> emergence of anticorrelations of 0.04; V1/V2 -> increased positive correlations of 0.04) and the matrix penalizing long-range connection (FEF -> emergence of anticorrelations of 0.03; V1/V2 -> increased positive correlations of 0.08) were all consistent with the original findings.

To address this legitimate concern, the following sentences have been added in the section (‘Anatomical connectivity’):

“Probabilistic tractography combined with the relatively large sample size allowed an accurate estimation of the average connectivity strength of pairwise regional connections in the adult population. However, to ensure that the findings of our model were robust to possible inaccuracies in streamline reconstruction, we ran the model using (i) a sparser structural connectivity matrix (30% density vs. original densely connected matrix), and (ii) a correction for the reconstruction of long-range connections (division of the original weights by the fiber distance between any two regions). These analyses replicated the original result showing that TMS of FEF increases the discrepancy between activity in this brain region and V1/V2 whereas TMS of V1/V2 decreases the discrepancy. This robustness is largely explained by the fact that FEF consistently has a high degree (i.e., is a hub) whereas V1/V2 is consistently a peripheral node with a comparatively low degree.”

*4) The Reviewing Editor raised a concern about the asymmetry in connectivity between FEF and V1/V2 at baseline and the other reviewers agreed this should be addressed. Specifically, the data in Figure 2 on baseline resting-state connectivity showing that a V1/V2 seed is negatively correlated with FEF while an FEF seed did not show a negative correlation with V1/V2. One would expect these effects to be roughly symmetric and it would be helpful to clarify why they are not. One possibility is that it's just a thresholding issue (e.g., 2b would show V1/V2 at a slightly more liberal threshold). Another possibility is that the seeds are not exactly in the same location as the sites that are correlated with the opposite seed). Please clarify.*

The Reviewing Editor is correct in expecting that the correlation between two regions should be roughly symmetric (i.e., functional connectivity between A and B should match that between B and A). The lack of a precise matching between the two is likely due to two factors. First, we performed a seed-to-voxel functional connectivity analysis and not a seed to seed (i.e., region-of-interest) analysis. This means that the clusters of voxels in FEF (V1/V2 seed analysis) and early visual cortex (FEF seed analysis) were similar but not identical. The absence of a significant anticorrelation between the FEF seed and early visual cortex at baseline may therefore be due to the fact that the FEF seed (red in the figure below) is located slightly laterally and posteriorly relative to voxels showing the highest anticorrelations with the V1/V2 seed (light blue in the figure below, Z= 50 as per Figure 2). Second, while the topology of the two baselines (V1/V2 and FEF sessions) was almost identical (see new Figure 2—figure supplement 2), anticorrelations between V1/V2 and FEF in the FEF baseline did not quite reach statistical significance.

*Recommended considerations*

*5) Two reviewers agreed the manuscript would be strengthened by the inclusion of control sites in the connectivity analysis while a third reviewer disagreed. The most useful additional analysis that the majority agreed would be worth investigating would be to examine the connectivity of the V1/V2 site after FEF stimulation and vice versa. We suggest the authors check this out and see if it adds any value to the paper (and include it if it does or provide a brief summary in the reply letter if it doesn't).*

The transitivity of the correlation coefficient as a measure of connectivity (e.g., Zalesky et al., Neuroimage, 2012) limits nodes which we can use as potential nulls. Because functional connectivity with V1 changes upon TMS of FEF, the Pearson’s correlation will change between V1 and all other regions showing a change of functional connectivity with FEF. Therefore V1 cannot be used as a null region. To address the reviewers’ comment with this caveat in mind, we conducted two new sets of analyses:

First, we assessed changes in functional connectivity pre- and post-TMS (for both V1/V2 and FEF sessions) using a control region ipsilateral to the TMS stimulation site but outside the networks of interest (here, the inferior portion of the right motor cortex, x= 51, y= -10, z= 18). These analyses revealed no significant changes in functional connectivity (p< 0.05 FWE corrected at cluster level) between the pre- and post-TMS scans (for both V1/V2 and FEF sessions). The following sentences have been added at the end of p. 7:

“We also assessed changes in functional connectivity pre- and post-TMS (for both V1/V2 and FEF sessions) using a control region ipsilateral to the TMS stimulation site but outside the networks of interest (i.e., the inferior portion of the right motor cortex, x= 51, y= -10, z= 18). These analyses revealed no significant changes in functional connectivity between the pre- and post-TMS scans, for both V1/V2 and FEF sessions.”

Second, we undertook two new *simulation* analyses. These analyses were undertaken to exploit the potential of the model to undertake “virtual” TMS and hence examine whether our functional connectivity results could arise from ‘non-specific’ effects of local stimulation. To this end, we examined whether:

(i) Simulated inhibition of the right postcentral gyrus (a region within the sensorimotor network, Power et al., Neuron, 2011) and left superior temporal gyrus (a region within the auditory network, Power et al., Neuron, 2011) would result in changes in functional connectivity similar to those observed following simulated TMS of V1/V2 and FEF;

(ii) Simulated inhibition of the right postcentral gyrus and left superior temporal gyrus could result in specific changes in functional connectivity between early visual cortices and FEF.

These simulations demonstrated that inhibition of either of these two control regions was associated with markedly different changes in functional connectivity compared with those observed following inhibition of V1/V2 and FEF (see new Figure 4—figure supplement 1, brains on the left side). Moreover, functional connectivity between V1/V2 and FEF was not significantly modulated by inhibition of either control site (new Figure 4—figure supplement 1, right side). Accordingly, the following sentences have been added to the revised manuscript (last paragraph of the Results section):

“To test whether the opposing changes in functional connectivity revealed by the simulations were driven by non-specific effects of local stimulation, we performed several additional simulations. First, we assessed whether inhibition (δ ω of 0.001) of cortical regions not relevant for our hypotheses – in this instance areas in the vicinity of the right postcentral gyrus and left superior temporal gyrus – could also produce effects on functional connectivity similar to those observed after simulated inhibition of V1/V2 or FEF. Second, we tested whether inhibition of these two control regions would cause specific changes in functional connectivity between early visual cortex and FEF. Simulated inhibition of either control region yielded a markedly different change in connectivity than that observed following simulated inhibition of V1/V2 or FEF (see Figure 4—figure supplement 1). Moreover, connectivity between V1/V2 and FEF was not affected by simulated inhibition of either of the two control regions.”

Reviewer #1:

*This paper presents a combination of TMS with different analyses of functional/effective connectivity and computational modelling, using a whole-brain formulation based on the Kuramato model. This combination is so far unique and enables the authors to provide evidence for a plausible and quite fundamental principle of cortical organisation, i.e., that cortical areas at different hierarchical levels operate at different time scales. I think this is a strong and innovative paper which elegantly combines complementary techniques. However, from my perspective, a few issues would benefit from clarification or reformulation:*

*1) I think the presentation of the key findings and the conclusions drawn needs to be improved. In brief, the key conclusion seems to be: inhibitory TMS leads to a local slowing of the timescale (frequency of dominant oscillations) at which an area operates; TMS of V1/V2 decreases and TMS of FEF increases the discrepancy in oscillatory frequencies between these two levels of the visual hierarchy; this explains why the former leads to increased bottom-up connectivity and the latter leads to reduced top-down connectivity. The authors get to this on fourth paragraph in the Discussion, but do not make it as explicit as I think they should.*

In response to the reviewer’s comment the following sentences have been added:

Final paragraph of Introduction:

“Experimental and modelling results support the notion that local inhibition of early visual cortex reduces the discrepancy in endogenous synchronization between lower and higher levels of the visual cortical hierarchy, whereas inhibition of FEF increases the discrepancy. Our work therefore provides novel insights into the neural mechanisms that underlie the effects of local inhibition on large-scale brain dynamics.”

Fourth paragraph of Discussion:

“The observed changes in connectivity following inhibitory TMS may therefore be explained in terms of a reduction in synchronization discrepancy following local stimulation of V1/V2 and an increase in synchronization discrepancy following stimulation of FEF.”

Moreover, it would help if this interpretation and the underlying hypothesis would be pointed out more clearly earlier in the paper, e.g. in the Introduction.

In keeping with our previous response, two sentences have been added at the end of the Introduction. This change also complies with the journal style of outlining the main results, and significance, at the end of the Introduction.

*2) To support this conclusion, it would be helpful if not only the correlation between changes in ALFF and changes in connectivity were shown; additionally, it would be important to see the magnitude and direction of the local changes in ALFF, separately at both levels of the hierarchy, that are induced by TMS.*

We now provide a table of the individual ALFF values for each participant as a function of targeted region (i.e., V1/V2 and FEF) and condition (i.e., baseline, post-TMS, difference baseline *minus* post-TMS) – see new [Supplementary-material SD2-data]. This complements the changes now visualized in Figure 3—figure supplement 4.

*3) One potential concern for the whole-brain Kuramato model is the accuracy of the structural connectome on which the model rests. This connectome was derived from diffusion-weighted imaging data, using probabilistic tractography. It would be helpful if the authors could provide some reassurance that these connectivity estimates are robust and not overly affected by methodological problems of tractography, such as the known bias in reconstructing short- versus long-distance connections.*

Please see our response to the 3^rd^ main comment (Essential changes) for a comprehensive treatment of this issue.

*4) Testing the robustness of the functional connectivity results across a variety of preprocessing strategies for resting state data is a strong aspect of this paper. However, it did not entirely get clear what motion correction procedure the results eventually reported are based on. It would be helpful if this could be clarified.*

We have now clarified this in the Materials and methods (Eighth paragraph):

“Thus, the results reported reflect the outcomes following motion correction (i.e.,re-alignment, regression of the six head motion parameters and scrubbing).”

Reviewer #2:

*The aim of the study was to explore long-range connectivity changes (assessed by resting-state fMRI) resulting from a local decrease in excitability induced by continuous theta burst TMS (cTBS). The authors used cTBS to modulate excitability in the V1 and the R-FEF in separate sessions. Inhibition of V1 with cTBS led to an increase in functional connectivity between V1 and R-FEF. Inhibition of R-FEF led to a decrease in functional connectivity between R-FEF and V1. This is an interesting study that looks to combine brain stimulation and neuroimaging with modelling to address a clearly articulated question.*

*Major Comments:*

*1) The authors suggest that there is a significant relationship between the amplitude of local BOLD signal within the FEF and changes in connectivity between FEF and V1 (Figure 3—figure supplement 3). Could the connectivity changes therefore be (trivially) explained by merely increased or decreased signal-to-noise in the seed region leading to changes in the connectivity? (If there is greater signal – and therefore greater variability – in the seed region, it is more likely that functional connectivity can be identified between that region and elsewhere).*

*This is a difficult problem to address, and I admit I do not have an easy solution, but it should be at the least discussed. I admit that I do not understand the sentence "the effects remained after controlling for changes in mean BOLD signal between baseline and post-TMS sessions", which may be aimed at addressing this conflict.*

While we understand the reviewer’s concern, here we argue that such an explanation is unlikely to account for the specificity and directionality of our results. First, we have shown that local changes in ALFF – following an identical stimulation protocol – are correlated with changes in functional connectivity and that these changes occur in opposite directions for the two TMS targets. That is, while a TMS-induced reduction in local ALFF correlated with the emergence of positive correlations with FEF following V1/V2 stimulation, a TMS-induced reduction in local ALFF correlated with the appearance of significant *anticorrelations* following stimulation of the FEF. Second, the fact that the correlations remained comparable after controlling for changes in mean BOLD signal across sessions further limits the likelihood that such correlations were simply driven by general fluctuations in SNR. Finally, we also examined activity within homologous visual areas in the opposite (non-stimulated) hemisphere. There were *no* significant correlations between changes in functional connectivity and changes in the amplitudes of slow signal fluctuations (ALFF values) extracted from homologous visual areas in the left hemisphere.

*2) The authors present no controls for their study. There are a number of questions raised by this:*

*a) Can they demonstrate that the fluctuations in connectivity between V1 and FEF, separated in time as these scans were, cannot explain the changes here (i.e. is this just an effect of repeated rs-fMRI scanning, rather than a TMS effect)? It may be possible to get this data from existing datasets – a step towards this would be to study the differences between the two baseline sessions, though this would not account for changes due to differing time spent in the scanner.*

*b) Are these changes specific to the stimulated regions? If the seeds for the functional connectivity analyses are placed elsewhere in the functional networks, can similar changes be elicited? And similarly for regions outside the networks? The authors have the data to perform these analyses.*

*c) Are the changes seen specific to stimulation at these sites? If the network was stimulated at a different site, would this result in the same pattern of results (i.e. is this just a reflection of some perturbation to the network as a whole or does it reflect the specific connectivity between these regions). This would require more data to be acquired, but is important for their conclusion that these are specific effects. A step towards this might be to show that stimulation of the FEF does not lead to connectivity changes in the V1 seed and vice versa, which they have the data to demonstrate.*

Please refer to our response to the above *“Recommended considerations”*. Our extensive new control analyses, on real and simulated data, directly address the reviewer’s concerns.

*3) One of the major things I struggled with in this study is the authors' interpretation of ante-correlations. To me, the demonstration of a significant, negative functional connection between two regions is not a lack of a functional relationship, a point with which the authors seem to agree. Therefore, a numerical increase in the r value between two regions is not an increase in connectivity if it does not go through 0, but rather is a decrease in (inhibitory) connectivity. If it does go through 0 it is then a reversal of negative to positive functional connectivity and the interpretation is very different.*

*This is an important point of interpretation, and is currently not clear in the manuscript. If I understand correctly, the authors suggest that inhibitory TMS to V1 led to an increase in connectivity between V1 and FEF. These regions were negatively correlated at baseline (Figure 2). Does this increase in connectivity mean that these regions are now positively correlated or that they are less negatively correlated? In the caption to Figure 3 they state that these are "antecorrelations" but I do not see the data to support that.*

*Likewise, inhibitory TMS to the right FEF led to a significant decrease in functional connectivity between this region and V1. Does this mean that there is now a significant functional connectivity between these regions, where there was not previously?*

Several sections of the manuscript have been revised accordingly; please see our response to the first comment in the section “*Essential changes”*.

*Reviewer #3:*

*The authors combine fMRI, TMS, and computational modeling to investigate interactions between visual cortical regions. They find that cortical stimulation of posterior visual cortex (V1/V2) leads to an increase in BOLD correlations with FEF, but stimulation of FEF leads to a decrease in BOLD correlations with posterior cortex. The modeling work indicates that these changes are related to the intrinsic timescales of these regions. This is a truly impressive study. The multi-methodology approach and research question is very novel and the results have important implications on the neural processes facilitating interactions along the cortical hierarchy. The manuscript is well written. Several aspects of the data that could affect the interpretation of the results need clarification. Specifically, the authors should evaluate potential influences of the underlying functional organization on the observed interactions, and address the differences between the imaging and modeling results. In addition, I have concerns on the specificity of the localization for both the imaging data and TMS. Conceptually, the authors also need to better address the relation of positive/negative BOLD correlations to increased and decreased connectivity.*

*Main Comments:*

*Positive and negative correlations are not synonymous with increased and decreased connectivity. The authors summarize their findings as "stimulation of early visual cortex selectively increased feedforward interactions with FEF" and " stimulation of FEF decreased feedback interactions with early visual areas." TMS of FEF resulted in a stronger anti-correlations with V1/V2. An argument could easily be made that connectivity increased in both cases, especially since these areas appear to be moderately anti-correlated at rest (see below).*

We agree with the Reviewer’s comment and have changed the text accordingly. Please refer to our response to the first comment in the “*Essential changes*” section.

*Are the observed interactions between V1/V2 and FEF generalizable to posterior visual vs. higher-order (frontal) regions? The modeling certainly appears to predict generalizable effects, but I'm not sure the imaging data specifically demonstrate this.*

In the original analysis (Figure 3) broader changes did not quite reach significance. In fact, inspection of the experimental data (Figure 4—figure supplement 3) suggests that sub-threshold changes in connectivity patterns occurred in regions comprising the frontal-prefrontal cortices and visual areas in the occipital cortex (as predicted by the model).

*Could the correlation patterns be related to the underlying topographic organization? Anatomical and functional studies in monkeys have shown that FEF and visual cortex is topographically organized with foveal and peripheral V1/V2 connected to differentiable portions of FEF (Schall et al. 1995; also see Babapoor-Farrokhran et al. 2013; Janssens et al. 2013). The particular increase/decrease in BOLD co-fluctuations may reflect where stimulation was applied relative to the functional organization of these regions (specifically retinotopy). This would still be an interesting result, but should be clarified.*

The proximity of the FEF stimulation sites and the relatively sparse effect of TMS precludes any unequivocal assessment of distinct changes between FEF sub-regions and foveal/peripheral cortical areas. We did, however, endeavor to address the reviewer’s query by testing for possible associations between the anatomical location of the stimulated sites and the broad changes in local neural activity (ALFF) and functional connectivity following TMS (new Figure 3—figure supplement 3, see below). These additional analyses failed to establish any clear link between the locations of the TMS sites, changes in local signal power (ALFF) and distal changes in functional connectivity.

In addition to the new Figure 3—figure supplement 3, the following sentences have been added in the manuscript:

Results section, subsection “Effects of local inhibitory TMS on functional connectivity”:

“Note that the individual sites of simulation could not be unequivocally linked with specific effects on functional connectivity (Figure 3—figure supplement 3).”

Results section, subsection “Impact of variability in the local response to TMS on group-level connectivity”:

Finally, there were no clear associations between changes in ALFF and individual TMS target sites (Figure 3—figure supplement 3).

*Given that TMS to FEF results in decreased BOLD activity in foveal V1, but increased BOLD activity in peripheral of V1 (Ruff et al. 2006; Ruff et al. 2009; Driver et al. 2010), were increases in BOLD correlations observed in peripheral V1/V2 for Post-FEF-TMS?*

We did not find this to be the case in our data (i.e., FEF stimulation did not cause the emergence of significantly positive correlations with V1/V2). This result could be due to the fact that we assessed resting-state dynamics and not task-based dynamics (for comments on state dependent effects of brain stimulation see Silvanto and Muggleton, NeuroImage, 2008, Silvanto et al. TICS, 2008; for recent reviews see Sale et al., Neurosci Biobehav Rev., 2015 and Miniussi et al., Handbook Clin Neurol., 2013).

Were any decreases in correlations observed in frontal cortex for POST-V1-TMS? If so, would this be predicted by the model?

No. We have edited the revised manuscript to state (2^nd^ paragraph p.7):

“There were no significant decreased correlations between V1/V2 and any cortical region following TMS of V1/V2.”

*How precise was the TMS? Judging by the provided MNI coordinates, it's not clear that the closest gyrus on the surface is still within V1/V2. It's very close to lateral occipital extrastriate areas such as LO1/2.*

As requested, we have now provided a figure showing the locations of the stimulation sites in each participant, in standard brain space (Figure 1—figure supplement 1).

A closer analysis of the TMS sites in relation to the known functional and structural definition of extrastriate visual areas indicates that we indeed targeted a region comprising areas V1/V2 (e.g., see figure below, A= modified figure from Malikovic et al., Brain Function and Structure, 2016). This panel shows the position and extent of cytoarchitectonically defined visual areas in the right hemisphere of ten human brains, B= TMS sites used in our study).

Given the lateral/medial distinction of FEF in the macaque, is it possible that TMS stimulation of FEF targeted a more foveal region than the seed region for the BOLD correlations?

Our results do show a relatively widespread effect of FEF stimulation on V1, V2 and other extrastriate visual areas (Figure 3). This lack of specificity is most likely due to a relatively broad perturbation of FEF. In fact, both lateral and medial FEF regions are likely to be affected by stimulation.

*The authors should address sources of variability in the localization of areas. Closer evaluation of the correlation maps could potentially address this. While the pre- and post- TMS correlation maps of FEF heavily overlap in the ipsilateral hemisphere (Figure 3, supplement 1), there is some separation with the post-TMS increase located slightly more inferior and lateral. Slices end at z = 40 in this figure though. Does the increase in correlation post TMS extend further inferior? What is the relationship between the observed correlations in FEF and the seed FEF?*

To follow up on the reviewer’s comment we investigated TMS-induced changes in FEF. The right cluster extends slightly inferiorly (up to Z = 36) but does not extend further backward. The following sentence has been added in the legend for Figure 3—figure supplement 1:

“Note that the FEF cluster showing positive correlations with V1/V2 following V1/V2 stimulation extends slightly inferiorly (up to Z = 36).”

*The model predicts that TMS to V1/V2 leads to a general increase in connectivity across cortex, and TMS to FEF leads to a general decrease. Can the model speak to potential topographically specific interactions as noted above?*

Preliminary investigations suggest that there are a host of topographical specificities and nuances that accompany virtual simulations. However, undertaking such work, and extracting underlying principles introduces substantial computational complexities that mandate a complete study of its own. To highlight this we have added the following sentence in the Discussion (Fourth paragraph of Discussion):

“Within this broad context, preliminary analyses suggest that there are a host of topographical specificities and nuances that accompany virtual simulations. Further work is needed to address the specificity of our model and the source of possible discrepancies between virtual and empirical findings.”

*What accounts for the differences between the model predictions (Figure 4) and the observed changes in BOLD correlations (Figure 3) post-TMS? Specifically, the model predicts increases in extrastriate cortex, temporal lobe, and much of the frontal lobe after TMS to V1 while changes in BOLD correlation are minimal in non-FEF frontal cortex, the temporal lobe, and extrastriate cortex (with the exception of a surprisingly specific increase in posterior parahippocampal cortex). There also appear to be increases in BOLD correlations within the parieto-occipital sulcus and retrosplenial cortex that are not clearly predicted in the model. After TMS to FEF, the model predicts large decreases in the frontal lobe and anterior, lateral occipital cortex (possibly near area MT) and increases in anterior temporal cortex, which are not apparent in the imaging data.*

Several factors might explain the discrepancies between the threshold results of the modelling and the experiment. In general, the model represents a considerable simplification of the underlying complex neuronal processes. Thus, the correspondence between the computational model and the empirical findings is necessarily limited. For example, empirical data are limited and thus underlying effects may not reach statistical thresholds. There are likely also considerable regional heterogeneities and related complexities that are not incorporated in the model. We have discussed the need to address possible discrepancies between model and data with the aforementioned edits to the Discussion (see above).

Reviewing Editor:

*I was puzzled by the data in Figure 2 on baseline resting-state connectivity showing that a V1/V2 seed is negatively correlated with FEF while an FEF seed did not show a negative correlation with V1/V2. One would expect these effects to be roughly symmetric and it would be helpful to clarify why they are not. One possibility is that it's just a thresholding issue (e.g., 2b would show V1/V2 at a slightly more liberal threshold). Another possibility is that the seeds are not exactly in the same location as the sites that are correlated with the opposite seed). Please clarify.*

Please refer to our response to the fourth comment under “*Essential changes”.*

[Editors' note: further revisions were requested prior to acceptance, as described below.]

*Essential points*

*1) The manuscript has become clearer by rephrasing in terms of changes to positive and negative correlations (rather than just increases or decreases without respect to the initial sign of the correlation as in the first version). However, both Reviewer #3 and the Reviewing Editor still found it hard to interpret without an additional figure to show the post-TMS connectivity. (not just pre and post-pre changes). See their comments for details (Reviewer 3, Point 1; RE, Point 1).*

To address this issue, we extracted single-subject average baseline values of functional connectivity in key clusters showing a significant TMS effect. We then calculated group-level mean connectivity, for both baseline and post-TMS clusters. These results are now presented in a new Table ([Supplementary-material SD2-data],). They allow a quantitative assessment of the direction of changes in functional connectivity induced by TMS.

Accordingly, we have modified the manuscript as follows:

Seventh paragraph of Results:

“Inhibitory TMS of right visual cortex (V1/V2) resulted in the emergence of positive correlations between this region and bilateral FEF (Figure 3—figure supplement 1 and [Supplementary-material SD2-data]). TMS of the visual cortex also resulted in the emergence/increase of positive correlations between V1/V2 and extrastriate regions including the lingual gyri and lateral occipital cortex (p< 0.05 FWE; Figure 3 – red, details in [Supplementary-material SD1-data] and [Supplementary-material SD2-data]). […].On the other hand, inhibitory TMS of right FEF resulted in a reduction in positive correlations between the targeted FEF region and visual areas encompassing the bilateral fusiform and occipital gyri (Figure 3 – blue, details in [Supplementary-material SD1-data] and [Supplementary-material SD2-data]).”

Results section, subsection “Computational modeling”:

“Conversely, reduced positive correlations following FEF stimulation were driven by a reduction in feedback connectivity (i.e., a reduced influence of FEF on V1/V2).”

Fourth paragraph of Discussion section:

“In contrast to the increased positive coupling apparent after inhibition of early visual cortex, an identical inhibitory TMS protocol delivered over the right FEF led to a decoupling between the target region and bilateral early visual cortex.”

Legend of Figure 3:

“Figure 3. Distinct effects of local inhibitory TMS over early visual cortex and FEF. (a) Inhibitory TMS of early visual cortex (right occipital pole; encircled lightning symbol) was associated with the emergence of positive correlations between BOLD signals in V1/V2 and bilateral FEF, and the emergence/increase of positive correlations between the V1/V2 seed region and bilateral occipital and parietal cortices (see [Supplementary-material SD1-data] and [Supplementary-material SD2-data] for details, p< 0.05 FWE corrected at cluster level). (b) Inhibitory TMS of the right FEF (encircled lightning symbol) resulted in a reduction of positive correlations between this target region and bilateral occipital visual areas (p< 0.05 FWE corrected at cluster level).”

Legend of Figure 3—figure supplement 1:

“Overlap of baseline and post-TMS connectivity between early visual cortex and FEF, bilaterally. Analysis revealed frontal regions that were anti-correlated with BOLD signal activity in the right occipital pole at baseline (illustrated in blue), regions that increased their connectivity (i.e., emergence of positive correlations) with right early visual cortex following inhibitory TMS (red-orange), and regions that were both anti-correlated with the right occipital pole and increased their functional connectivity with this region following TMS (pink). Note that the FEF cluster showing positive correlations with V1/V2 following V1/V2 stimulation extends slightly inferiorly (up to Z = 36). All p< 0.05 FWE, cluster level.”

*2) Although based on your reply, we understand your justification for displaying data on the folded cortical surfaces, the problem remains that one of your two key areas (FEF) is hardly visible from lateral views. This can be solved simply by presenting either a superior view or a horizontal slice consistently in all figures (not just a subset as it is currently, e.g., slice on Figure 2, superior view on Figure 3). And you may as well add an inferior view (to show lingual/fusiform cortex).*

As requested we have updated Figure 2, Figure 2—figure supplement 1 and Figure 2—figure supplement 2. These figures now present both the superior and inferior views.

*3) In the post-review consultation, the other reviewers reinforced a suggestion from Reviewer #3 (last line of Reviewer 3, Point 1). Reviewer #1 stated: I think the one issue to resolve is whether the model does or does not predict anti-correlations, that is, the authors should clarify the connection between their description in the main text and the results shown by Figure 4.*

To address this comment we have added a new supplementary figure (Figure 4—figure supplement 1) showing that the model predicts both the reduction of positive correlations and the emergence of anticorrelations between the right FEF and right V1/V2 following inhibitory TMS of FEF.

*Recommended considerations*

*Reviewer #3 raised two points that the other reviewers (in post-review consultation) were less convinced were essential to resolve. Nevertheless, the Reviewing Editor would encourage you to take this feedback into consideration in case you can use it to strengthen the manuscript.*

*1) Reviewer #3 (Point 2) still questions the asymmetry of the FEF and V1/V2 correlations. The Reviewing Editor thinks that if it cannot be fully resolved, it should at least be made more apparent in the main manuscript (by showing the FEF clearly in extant figures and/or moving Figure 3—figure supplement 1 to the main text) and discussed. As is, the main figures don't show this because of the views presented).*

See our response to Q2. (Essential Changes) above.